# *Representation Projection Invariance*
# Mitigates Representation Collapse

**Anastasia Razdaibiedina**◇ **Ashish Khetan**♣ **Zohar Karnin**♣
**Daniel Khashabi**♠ **Vishaal Kapoor**♣ **Vivek Madan**♣
◇University of Toronto ♠Johns Hopkins University ♣Amazon AI
anastasia.razdaibiedina@mail.utoronto.ca danielk@jhu.edu
{vivmadan, zkarnin, khetan, vishaalk}@amazon.com

## Abstract

Fine-tuning contextualized representations learned by pre-trained language models remains a prevalent practice in NLP. However, fine-tuning can lead to *representation degradation* (also known as *representation collapse*), which may result in instability, sub-optimal performance, and weak generalization.

In this paper, we propose *Representation Projection Invariance* (REPINA), a novel regularization method to maintain information content of representation and reduce representation collapse during fine-tuning by discouraging undesirable changes in the representations. We study the empirical behavior of the proposed regularization in comparison to 5 comparable baselines across 13 language understanding tasks (GLUE benchmark and six additional datasets). When evaluating in-domain performance, REPINA consistently outperforms other baselines on most tasks (10 out of 13). Additionally, REPINA improves out-of-distribution performance. We also demonstrate its effectiveness in few-shot settings and robustness to label perturbation. As a by-product, we extend previous studies of representation collapse and propose several metrics to quantify it. Our empirical findings show that our approach is significantly more effective at mitigating representation collapse.[1]

## 1 Introduction

Fine-tuning pre-trained language models has been shown to achieve remarkable performance on a variety of natural language processing (NLP) tasks (Kenton and Toutanova, 2019; Brown et al., 2020; Zhang et al., 2022). A standard fine-tuning strategy involves adapting the pre-trained model to a supervised downstream task (Fig 1; left). Such a procedure can result in *representation collapse* (Aghajanyan et al., 2021; Zhou and Srikumar, 2022), distorting the pre-trained representations that limits

[1]Code is available at https://github.com/arazd/REPINA.

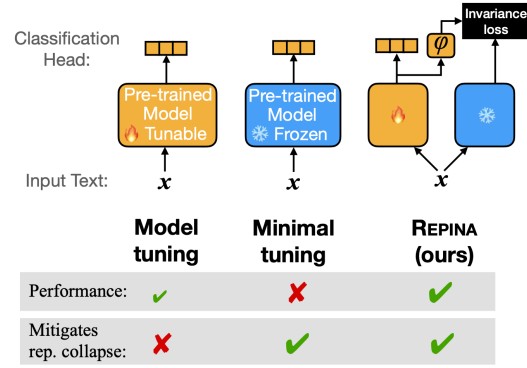

Figure 1: Fine-tuning the whole architecture (left) generally leads to good performance though it distorts the pre-trained representations. Minimal tuning (of classification head, for example; middle) mitigates the representation collapse but limits the model performance. Our proposal REPINA (right) leads to good performance while mitigating the representation collapse.

their generalizability to other domains, styles, or tasks. An alternative approach to full model tuning is to fine-tune only several top layers, while keeping the rest of the model frozen (e.g., we could train solely a classification head, Fig 1; middle). This practice of freezing all/most of the model parameters can prevent unwanted changes to pre-trained representations, but it can also limit fine-tuning and negatively affect performance (Lee et al., 2019b; Kumar et al., 2021). This study aims to determine if it is possible to fine-tune the entire model without compromising representation quality.

We introduce *Representation Projection Invariance* (REPINA), a regularization objective that prevents undesirable changes in the representations (Fig 2a). Our regularization applies an *invariance* loss on a tunable *projection* of the representation. This regularization allows the underlying representation to change mildly (e.g., shift and scaling) while not losing its expressivity (Fig 2b). Our regularization objective provides a knob that controls the amount of loss-free transformations allowed

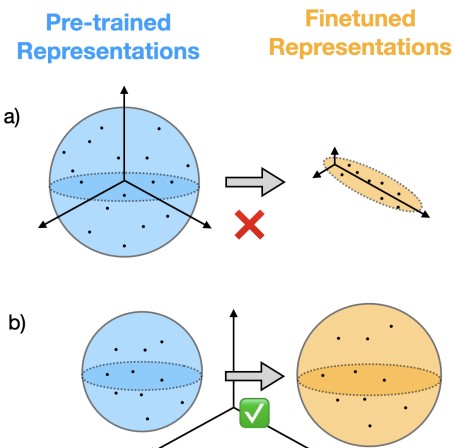

Figure 2: Example **a)** shows representations *collapsing* into a single dimension and losing useful information after fine-tuning. Example **b)** shows changes in representations that preserve their expressive power (e.g., coordinate shift, rotation, scaling, etc.).

during fine-tuning.

We compare our method against several established regularization approaches which explicitly or implicitly address the issue of representation degradation (Section 5.1). We show that our approach consistently outperforms major fine-tuning methods across 13 tasks (Fig 3; left) and improves generalizability on out-of-distribution (OOD) data (Section 5.4). Our approach is particularly effective in scenarios where data is limited (such as with only 250, 500, or 1000 examples), as the model is more likely to overfit and memorize the training data in these cases (Section 5.6). Furthermore, we thoroughly investigate fine-tuning under label perturbation (from 5% to 30% label noise) and observe that our approach is robust to incorrect labels, exceeding the performance of standard fine-tuning procedure and common baseline methods (Section 5.5).

Finally, we quantify how much different methods mitigate the degradation of representations (Section 6). We use previously explored probing experiments (Aghajanyan et al., 2021), and propose a new set of metrics that quantify representation collapse objectively, without requiring extra datasets/training. We observe that REPINA shows the strongest resistance to representation degradation among all methods.

## 2 REPINA: Representation Projection Invariance

Our method avoids representation collapse by preventing undesirable changes in representations dur-

ing the fine-tuning process. A straightforward implementation would anchor representations during fine-tuning to their pre-trained values. That is, the final loss $\hat{\mathcal{L}}$ would combine the standard fine-tuning objective and regularizer of the deviation in representations:

$$\hat{\mathcal{L}} = \mathcal{L} + \lambda \sum_{x \in \mathcal{I}} \|f_{pre}(x) - f_{fin}(x)\|_2^2, \quad (1)$$

where $\mathcal{L}$ is the downstream task loss (e.g., cross entropy for classification tasks), $\mathcal{I}$ are the input samples of the task, $f_{pre}$ and $f_{fin}$ are the representation functions defined by the pre-trained and fine-tuned networks. Optimizing full model parameters under this modified objective would prevent representation degradation. However, this formulation of the loss function could be very restrictive.

Various transformations of a representation maintain its expressivity (such as linear shift; Fig 2b). While such transformations do not change the information content of a representation, they incur a high regularization loss based on equation 1. To address this issue and allow flexibility in representations while preserving their expressive capacity, we propose **re**presentation **p**rojection **in**variance regularization (REPINA):

$$\hat{\mathcal{R}} = \min_{\phi \in \Phi} \sum_{x \in \mathcal{I}} \|f_{pre}(x) - \phi(f_{fin}(x))\|_2^2$$
$$\hat{\mathcal{L}} = \mathcal{L} + \lambda \hat{\mathcal{R}}. \quad (2)$$

Here $\Phi$ is a class of dimension preserving functions chosen before the fine-tuning process and defines the strength of regularization. The intuition behind the regularization objective is to incentivize the representations to be *invariant* under some *projection* $\phi \in \Phi$; pre-trained representations can be constructed from the fine-tuned representations by applying a function $\phi \in \Phi$. For instance, if $\Phi$ is the set of linear functions $\{\phi \mid \exists W, b : \phi(z) = Wz + b\}$, then we bias fine-tuned representations to be linear transformations of the fine-tuned representations. Thus, regularization loss in case of Fig 2a would be zero since there exists a linear mapping from fine-tuned representations to the pre-trained representations. However, regularization loss for Fig 2a would be high as there does not exist such a linear mapping from fine-tuned to the pre-trained representations.

**Choice of the class of functions $\Phi$:** $\Phi$ defines the strength of the regularizer. For instance, a singleton $\Phi$ containing an identity function is the strongest

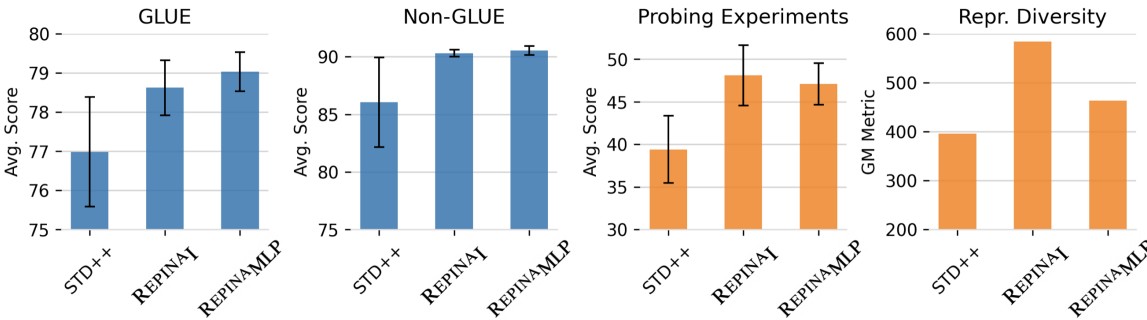

Figure 3: REPINA improves finetuning performance and mitigates representation collapse. STD++: Improved variant of standard finetuning. REPINA$_I$ and REPINA$_{MLP}$ are our methods. GLUE & Non-GLUE: Average test set performance across seven GLUE and six non-GLUE tasks. Probing Experiments: The measure of representation collapse introduced by Aghajanyan et al. (2020) (higher is better). Representation Diversity: Mathematical measure of the information content of representations (we report GM-5 score, see Section 6; higher is better).

regularizer which keeps fine-tuned representations close to the pre-trained representations (equivalent to equation 1). Conversely, for a rich $\Phi$, e.g., deep and wide neural networks, $\phi$ can be chosen to reconstruct the lost information in representation even if there is a severe degradation. Thus, it provides a weak regularization. Choice of $\Phi$ then depends on how prone fine-tuning is to overfitting and how strong of a regularization method is needed. For instance, few-shot setting may require the strongest regularization and larger training datasets may require milder regularization.

In this paper, we experiment with $\Phi$ containing identity function (REPINA$_I$ ) and shallow multilayer perceptrons (REPINA$_{MLP}$ ).

**Choosing the right representation:** Normally, sentence-level representations are obtained from the final encoder blocks of a transformer model. However, it may be more beneficial to use representations from the lower layers. In fact, Zhang et al. (2020) show that re-initializing weights of the top layers of encoder improves fine-tuning performance, suggesting that representation consistency may not be desirable for the top layers. Thus, we consider a variant of regularization that uses representations from the intermediate layers of encoder.

**Explaining representation invariance regularization as implicitly learning multiple tasks:** Consider the overfitting in Fig 2a again. It can be prevented by fine-tuning the representations for multiple tasks simultaneously instead of a single task. This multi-tasking is a well-known way to prevent overfitting. It not only prevents overfitting of representations but can also improves generalization performance for all of the tasks. We show that REPINA's regularization objective (equation 2) is equivalent to fine-tuning on multiple hypothetical tasks. Due to space constraints, we defer further

discussion on the connection and the formal proof of equivalence in Appendix B.

## 3   Related Work

**Mitigating Representation Collapse:** Aghajanyan et al. (2020) study representation collapse and propose two methods, *R3F* and *R4F*, to address it. *R3F* induces bias towards a solution with a locally smooth prediction function, and *R4F* extends *R3F* by adding a Lipschitzness constraint on the top classification layer. Some of the other methods which implicitly target representation collapse are *FreeLB* and *SMART*. *FreeLB* uses adversarial training to improve fine-tuning (Zhu et al., 2020) and *SMART* is a trust region-based method that avoids aggressive updates during fine-tuning (Jiang et al., 2020). *R3F* (Aghajanyan et al., 2020) has been shown to outperform both of these methods. Thus, we only include *R3F* in our set of baselines.

A method that specifically targets representations during fine-tuning is *Supervised Contrastive Learning* (SCL). SCL induces representations of examples with the same label to be close to each other and far from the examples of other classes (Gunel et al., 2020). A major disadvantage of SCL is a requirement for large mini-batch size and, hence, heavy memory consumption. We implement a memory-efficient SCL version but exclude the original implementation from the baselines due to computational cost (see Appendix H). Another method which can be potentially useful for mitigating representation collapse and form part of our baseline is *Data augmentation*. It can be done via back translation, synonymn replacement, random deletion, and synthetic noise and is known to improve generalization performance (Feng et al., 2021; Wei and Zou, 2019).

**Catastrophic Forgetting** (Kirkpatrick et al., 2017) is a phenomenon closely related to representation collapse. In a sequential training setting, it refers to forgetting information learnt from previous tasks while being trained on a new task. In our context, this means forgetting the pre-training language modeling task while training for the fine-tuning task. In contrast to catastrophic forgetting, we measure representation collapse as the loss in expressive power of the representations irrespective of the performance on pre-training task. A method known to alleviate catastrophic forgetting is *Weight Consolidation* (Chen et al., 2020; Kirkpatrick et al., 2017). It regularizes the fine-tuning process by encouraging fine-tuned weights to be close to the pre-trained weights. In contrast to weight consolidation, REPINA *does not put direct constraints on weight updates, but tries to control the structural changes in representations.*

Due to our limited space, we discuss further details on related works in Appendix A.

## 4 Experimental Set Up

### 4.1 Our Methods: REPINA$_I$ & REPINA$_{MLP}$

Recall our methods REPINA$_I$ and REPINA$_{MLP}$ introduced in Section 2. For REPINA$_I$ , we observe that regularizing intermediate layer representations (5th, 10th, 20th from input) perform better than regularizing the top layer (near the output, before the classifier) representation. Thus, the regularization objective for REPINA$_I$ is:

$$\hat{\mathcal{R}}(\Phi = \{\mathbb{1}\}) = \sum_{x \in \mathcal{I}} \|f_{pre}^{\ell}(x) - f_{fin}^{\ell}(x)\|_2^2,$$

where $f_{pre}^{\ell}, f_{fin}^{\ell}$ are $\ell$-th representations from the $\ell$-th layer (from input) of the model. Choice of $\ell$ is a hyper-parameter with possible values of $5, 10$ and $20$. Layer 5 is most effective for small training datasets and layer 20 is most effective for large training datasets (see Appendix E).

Due to computational limitations, we experiment with only top layer representations for REPINA$_{MLP}$ . Thus, the regularization loss for REPINA$_{MLP}$ , $\hat{\mathcal{R}}(\Phi = \text{MLP})$ is:

$$\min_{\Theta} \sum_{x \in \mathcal{I}} \|f_{pre}(x) - \text{MLP}_{\Theta}(f_{fin}(x))\|_2^2,$$

where $f_{pre}, f_{fin}$ are the representations from the top layer of the model (before the classifier) and $\Theta$ are the parameters of a multi-layer perceptron

(MLP). We set the depth of MLP to 2, keeping the width equal to the representation dimension. By varying the depth from 1 to 5, we observe that for smaller training datasets, lower depth performs better. Training with large datasets is robust to the depth choice (see Appendix D).

### 4.2 Baselines

We use a diverse range of baselines for our study:
**STD++** is an improved variant of the standard fine-tuning scheme that includes the use of bias correction in AdamW, following the works of (Zhang et al., 2020; Mosbach et al., 2020) which shows that bias correction is a major cause of instability in language model fine-tuning.

**Weight Consolidation** (Kirkpatrick et al., 2017; Chen et al., 2020) is an approach that encourages agreement between pre-trained $\theta^{pre}$ and fine-tuned $\theta^{fin}$ models weights via a regularization term $\sum_i \|\theta_i^{fin} - \theta_i^{pre}\|_2^2$ added to the loss function.

**R3F** (Aghajanyan et al., 2020) is a local smoothness regularization that prevents aggressive model updates by restricting divergence of outputs upon input perturbation. For model $f(\cdot)$ and input token embeddings $x$, R3F adds a regularization term $\text{KL}_S\left(f(x)\|f(x+z)\right)$ to the loss function, where $\text{KL}_S$ is the symmetric Kullback-Leibler divergence and noise $z$ is sampled from a normal distribution.

**ReInit** (Zhang et al., 2020) improves fine-tuning performance by re-initializing the top-$k$ layers of the encoder (closer to the output) with gaussian random samples from $\mathcal{N}(0, 0.02^2)$. Following the original study, we perform hyperparameter search for $k = 2, 4$ or $6$.

**Data Augmentation (DA)** generates augmented samples by adding noise to the training data (keeping the label intact) (DeVries and Taylor, 2017). In our implementation, we add gaussian noise $\epsilon \sim \mathcal{N}(0, \delta)$ to the token embeddings where $\delta = 1e-5$.

Table 2 show the regularization coefficients used for each method.

### 4.3 Datasets

We evaluate methods on GLUE benchmark (Wang et al., 2018) and six additional non-GLUE datasets (Table 3). These include: biomedical relation extraction on CHEMPROT (Kringelum et al., 2016), sentiment classification on YELP (Zhang et al., 2015a) and IMDB (Maas et al., 2011), citation intent classification on SCICITE (Cohan et al., 2019), language inference on SCITAIL (Khot

| Method ↓ / Task → | RTE | MNLI | SST2 | MRPC | QNLI | QQP | CoLA | Yelp | Chem | IMDB | AGnews | SciTail | SciCite |
|---|---|---|---|---|---|---|---|---|---|---|---|---|---|
| STD++ | 70.8 | 65.6 | 92.1 | 86.8 | 87.2 | 76.7 | 59.70 | 95.3 | 82.6 | 93.2 | 91.7 | 71.6 | 81.9 |
| DA | 73.6 | 65.5 | 92.0 | 90.7 | 87.4 | 76.4 | **63.4** | 95.6 | 82.9 | 93.2 | 91.8 | 93.7 | 82.1 |
| WC | 72.2 | **66.7** | 92.7 | 88.6 | 87.2 | 76.2 | 61.5 | 95.9 | **83.9** | 93.4 | 91.9 | 94.0 | 82.2 |
| ReInit | 70.9 | 65.1 | 92.0 | 91.0 | 87.3 | 77.2 | 61.2 | 95.4 | 82.5 | 92.7 | 91.7 | 93.4 | 82.4 |
| R3F | 70.4 | 65.0 | 92.1 | 89.9 | 87.0 | 74.9 | 62.0 | 95.5 | 82.9 | 93.1 | 91.7 | 86.5 | 82.0 |
| REPINA$_I$ | 71.4 | 65.7 | 92.9 | **91.5** | 87.5 | 79.0 | 62.3 | 95.8 | 83.5 | **94.0** | **92.1** | 93.7 | 82.7 |
| REPINA$_{MLP}$ | **74.4** | 65.2 | **93.2** | 91.1 | **87.6** | **79.3** | 62.5 | **96.0** | 83.7 | 93.9 | 91.9 | **94.8** | **83.2** |

Table 1: Performance for our methods (REPINA$_{I/MLP}$) and baselines on 7 GLUE and 6 non-GLUE datasets. Average gain of 2.1 over STD++ for GLUE datasets and 4.5 over non-GLUE datasets. REPINAbeats all baseline methods in 10/13 cases. For QQP, MNLI, QNI, AGNEWS, IMDB, YELP and SCITAIL, we only used 10K training datapoints.

| Method | Regularization coefficient |
|---|---|
| REPINA$_I$ | 0.01, 0.05, 0.1, 0.5 |
| REPINA$_{MLP}$ | 0.01, 0.05, 0.1, 0.5 |
| DA | 0.05, 0.1, 0.2, 0.4, 0.8 |
| R3F | 0.1, 0.5, 1, 5 |
| WC | 0.01, 0.05, 0.1, 0.5 |

Table 2: Regularization coefficient for different methods.

et al., 2018) and article topic classification on AG-NEWS (Zhang et al., 2015b). For each task, we use their corresponding adopted performance metric

On these 13 datasets, we conduct a variety of experiments with many and few supervision instances. To keep the cost of fine-tuning computations on extremely large datasets (such as MNLI and QQP), we limited their training sets to $10,000$ data points, and marked with a suffix "-10K" henceforth. For datasets with no available test set labels, we use their development set to report the performance. We use a subset of original train data split (size equal to validation set) which is not used for training for hyper-parameter selection.

| Task | Train | Dev | $C$ | Metric |
|---|---|---|---|---|
| COLA | 8551 | 1043 | 2 | MCC |
| RTE | 2490 | 277 | 2 | Accuracy |
| SST | 67349 | 872 | 2 | Accuracy |
| MNLI-10k | 10000 | 9815 | 3 | MCC |
| MRPC | 3668 | 408 | 2 | F1 |
| QQP-10k | 10000 | 40430 | 2 | F1 |
| QNLI-10k | 10000 | 5463 | 2 | Accuracy |
| CHEMPROT | 4169 | 2427 | 13 | Micro F1 |
| SCICITE | 7320 | 916 | 3 | Macro F1 |
| SCITAIL-10k | 10000 | 1304 | 2 | Accuracy |
| AGNEWS-10k | 10000 | 5000 | 4 | Macro F1 |
| YELP-10k | 10000 | 10000 | 2 | Accuracy |
| IMDB-10k | 10000 | 5000 | 2 | Macro F1 |

Table 3: The datasets used in this study, their size, number of classes ($C$) and the corresponding evaluation metrics. MCC denotes Matthews correlation coefficient.

## 4.4 Fine-tuning Settings

Due to the large scale of the experiments and in order to have a meaningful comparison with various approaches, we consistently use BERT-large model for implementing both our proposed algorithm and the baselines. Existing works such as (Zhang et al., 2020; Mosbach et al., 2020) also use similar experimental setups. Additionally, to verify the generality of our findings to other models, we performed limited experiments on RoBERTa-base where we observe similar performance gain.

We fine-tune all models for 5 epochs (unless otherwise specified) at a learning rate of 2e-5, and report performance with 5 different seeds. Due to resource constraints and in contrast to prior works (Kenton and Toutanova, 2019; Aghajanyan et al., 2020; Chen et al., 2020), we do not search for optimal learning rate for each method-task combination. To verify the impact of this choice, we perform limited experiments selecting the best learning rate, dropout and number of epochs for each method and a subset of tasks (Appendix F). We observe similar gains as reported in the main paper. For each method, we select optimal hyperparameters by performing evaluation on the unused fraction of the training set (see Appendix C).

Since standard fine-tuning is susceptible to failed runs that substantially lower the resulting performance (Mosbach et al., 2020; Razdaibiedina and Brechalov, 2023), we filter out failed runs and report average performance over 5 successful runs. We define run as failed if its performance is close to or lower than the majority classifier (i.e. a dummy model that always predicts the label of the majority class in the dataset) (Dodge et al., 2020). We define a threshold close to the performance of the majority classifier as per metric in Table 3. A fine-tuning run is "failed" if its performance on unused part of the training dataset is below the threshold. See Section C.2 for the exact thresholds.

## 5 Results: Generalization Performance

In this section, we present experimental results with the baselines introduced in the earlier section.

### 5.1 Full dataset - Generalization performance

Table 1 shows that **REPINA models outperform the baselines consistently across a variety of tasks**: our method outperforms other ones on 10/13 tasks. Both REPINA$_I$ and REPINA$_{MLP}$ outperform baseline methods in terms of mean performance, with improvements in the mean performance over the corrected fine-tuning strategy STD++ by 1.7 and 2.0 points, respectively, for GLUE benchmark, and 4.3 and 4.5 points for non-GLUE benchmark.

### 5.2 Analyses on Fine-tuning Stability

Similar to the prior literature (Dodge et al., 2020; Mosbach et al., 2020; Wang et al., 2018), we observe that the standard fine-tuning procedure is prone to instability and sub-optimal convergence, leading to failed runs. Recall that we formally define a fine-tuning run as a failed run if the resulting performance is close to the majority classifier.

In the previous section, we reported the mean performance of only successful runs (complement of failed runs). Figure 4 shows the fraction of runs that were successful for each method. We note that REPINA$_I$ has the least number of failed runs (maximum number of successful runs). Moreover, if we do not filter out failed runs, our methods perform even better than all the baseline methods. REPINA$_I$ achieves an average 2.6 percentage point improvement over the next best baseline method (WC). Thus, we conclude that **our methods demonstrate higher stability and less fraction of failed runs than other approaches**. (additional experiments in Table 21 in Appendix K.)

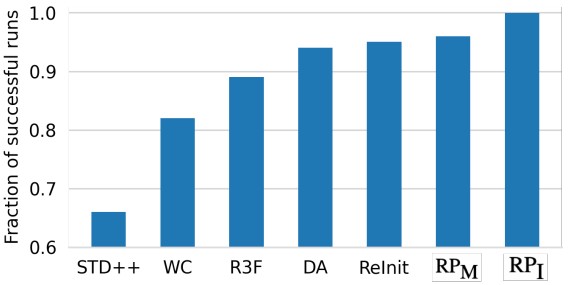

Figure 4: Fraction of successful runs across all tasks. Run is defined as successful if its test performance is higher than the performance of a majority classifier. Our proposed regularization (RP$_I$ /RP$_M$ ) increases the fraction of successful runs, hence, leading to more stable fine-tuning behavior.

### 5.3 Comparison with parameter-efficient tuning methods

The main focus of our study is mitigating representation collapse in a full model tuning setting, which is a common way to tune LLMs since it allows to achieve the best downstream task performance, at the cost of forgetting and representation collapse. However, this goal can also be achieved with parameter-efficient tuning (PEFT) methods, which can balance good downstream performance with minimal parameter changes.

As we have discussed in the introduction, if we fine-tune only several top layers of the model, while keep- ing the rest of the parameter frozen (e.g., train solely a classification head, Fig 1; middle), representation collapse can be avoided. However, this would not lead to optimal downstream task performance. As a middle ground approach, we could train just a larger fraction of model parameters (yet not the full model) with parameter-efficient approaches. Common PEFT methods include LoRA (Hu et al., 2021), prefix tuning (Li and Liang, 2021), prompt tuning (Lester et al., 2021), adapters (Houlsby et al., 2019), and variations of those (Rücklé et al., 2020; Khashabi et al., 2021; Lester et al., 2022; Razdaibiedina et al., 2023a,b).

We provide results for prompt tuning, prefix tuning, LoRA and REPINA-MLP on 4 GLUE tasks in Table 4. As we can see, **REPINA-MLP consistently outperforms PEFT methods**.

|  | SST-2 | MRPC | CoLA | RTE |
|---|---|---|---|---|
| Prompt Tuning | 86.4 | 76.9 | 59.9 | 52.5 |
| Prefix Tuning | 90.9 | 91.0 | 57.6 | 73.9 |
| LoRA | 91.5 | 90.0 | 60.4 | 71.5 |
| REPINA-MLP | **93.2** | **91.1** | **62.5** | **74.4** |

Table 4: Comparison with PEFT methods.

### 5.4 Out-of-distribution robustness

We perform experiments on four pairs of datasets to study OOD generalization following Hendrycks et al. (2020) protocol (see Table 5). The chosen pairs of datasets induce a distribution shift, which allows to measure OOD robustness. Due to limited resources and different sizes of tasks, we limited the training set to 200 examples per class (fixed across all runs). Overall, **REPINA$_I$ shows steady improvement in OOD performance in all cases**.

| Task | IID | | OOD | |
|---|---|---|---|---|
| Train → Eval | STD++ | RP$_I$ | STD++ | RP$_I$ |
| Imdb → SST2 | $92.1_{\pm 0.5}$ | $92.1_{\pm 0.5}$ | $87.8_{\pm 0.9}$ | $88.9_{\pm 0.5}$ |
| SST2 → Imdb | $91.1_{\pm 0.1}$ | $92.1_{\pm 0.1}$ | $87.1_{\pm 0.1}$ | $87.8_{\pm 0.1}$ |
| Yelp → Amzn | $60.7_{\pm 0.2}$ | $60.8_{\pm 0.9}$ | $28.9_{\pm 0.2}$ | $29.3_{\pm 0.5}$ |
| Amzn → Yelp | $39.4_{\pm 1.9}$ | $41.5_{\pm 1.4}$ | $49.7_{\pm 1.9}$ | $50.8_{\pm 1.4}$ |

Table 5: REPINA improves OOD performance. RP$_I$ : REPINA$_I$ ; Train: training task (IID), Eval: evaluation-only task (OOD). Data is limited to 200 samples/class.

## 5.5 Robustness to Label Perturbation

Real-world data can often contain mislabeled samples, which can hinder the training process. Hence, robustness to label noise is a desirable quality of the fine-tuning approaches. Here, we study the performance of the fine-tuning methods under label perturbation. We introduce **label noise** as follows: let $C = \{1, \ldots, c\}$ be a class of labels and $\mathcal{X} = \{(x, y)\}$ be the true dataset for the fine-tuning task. Our fine-tuning method has access to a noisy dataset $\mathcal{X}' = \{(x, y')\}$ where $y' = y$ with probability $1 - p$ and sampled uniformly from $\{1, \ldots, c\} \setminus \{y\}$ with probability $p$.

**REPINA$_I$ and REPINA$_{MLP}$ show the highest resistance to label perturbation**, retaining closest to the original performance upon introducing 5-10% noise to labels (Table 6). The second most resistant approach, WC, is also close to our method conceptually, as it discourages the fine-tuned weights to deviate from pre-trained weights.

| Noise ↓ | STD++ | DA | WC | ReInit | R3F | RP$_I$ | RP$_M$ |
|---|---|---|---|---|---|---|---|
| 0% | 64.7 | 78.5 | 81.4 | 79.9 | 72.9 | **84.0** | 83.0 |
| 5% | 58.3 | 68.2 | 75.3 | 72.3 | 57.3 | **81.4** | 78.0 |
| 10% | 58.0 | 63.7 | 72.2 | 68.9 | 52.4 | **78.1** | 75.6 |
| 20% | 48.4 | 49.1 | 64.1 | 55.2 | 44.3 | 66.2 | **70.2** |
| 30% | 40.1 | 45.9 | 53.5 | 52.4 | 42.0 | 50.3 | **59.5** |

Table 6: Mean performance over 13 datasets when training with noisy data. RP$_I$ : REPINA$_I$ , RP$_M$ : REPINA$_{MLP}$ . See Appendix L for detailed results.

## 5.6 Analyses on Few-shot Fine-tuning

To investigate our methods' robustness to small dataset sizes, we study REPINA$_{MLP}$ and REPINA$_I$ performance in limited data settings (250/500/1000 training data points). We fix the same data subset across all models to avoid performance changes related to data variability.

Since finetuning in few-shot setting is particularly prone to instability and the performance on a single dataset can distort the mean statistics for the entire collection, we use average rank as a more stable metric to compare different methods.

A method's rank for a task corresponds to the position of the method in a list of all methods sorted by performance on that dataset. The minimal and best possible rank is 1. The *average rank* of a method is obtained by averaging ranks across all tasks.

We observe in Table 7 that **REPINA$_I$ is the most effective method in the few-shot setting measured in terms of the average rank**. See Appendix J for a detailed analysis.

| # samples ↓ | STD++ | DA | WC | ReInit | R3F | RP$_I$ | RP$_M$ |
|---|---|---|---|---|---|---|---|
| 250 | 5.62 | 4.92 | 4.62 | 3.00 | 4.04 | **2.50** | 3.31 |
| 500 | 6.08 | 4.38 | 3.69 | 3.31 | 4.85 | **2.77** | 2.92 |
| 1000 | 5.69 | 4.00 | 3.62 | 3.54 | 4.62 | **2.69** | 3.85 |

Table 7: Average rank of different methods for few-shot learning. RP$_I$ : REPINA$_I$ , RP$_M$ : REPINA$_{MLP}$ .

Overall, we find that REPINA$_{MLP}$ yields performance gain on large-scale datasets, whereas REPINA$_I$ is effective for few-sample fine-tuning (since newly introduced parameters in REPINA$_{MLP}$ are undertrained when the training data is limited). For wall-time analysis, see Appendix O. For experiments on hyper-parameter optimization over learning rate, batch size and other hyper-parameters see Appendix F.

## 6 Degradation of Representations: Analytical Perspective

Here we quantify the representation collapse.

### 6.1 Probing Representation Collapse

We follow the setting of Aghajanyan et al. (2020) for studying representation collapse with *probing experiments* as follows: (i) fine-tune model on a downstream task $A$, (ii) freeze the encoding layers and train the top linear layer for a different task $B$. Low performance in the second step implies representation collapse in the first step. To assess robustness of the proposed approach to representation collapse, we perform a series of probing experiments. In our experiments, we use four GLUE and four non-GLUE datasets in the first step and all datasets in the second step except the one used in the first step (Table 8).

We observe that REPINA$_{MLP}$ and REPINA$_I$ show high resistance to representation collapse, outperforming other approaches in 6/8 cases (Table 8). For instance, fine-tuning for QNLI-10k in the first step with REPINA$_{MLP}$ results in a mean performance of 49.5 in the second step, whereas the next best baseline results in a mean performance of 44.5.

| Task A ↓ | STD++ | DA | WC | ReInit | R3F | RP$_I$ | RP$_M$ |
|---|---|---|---|---|---|---|---|
| QNLI | 37.6 | 37.1 | 44.5 | 37.7 | 36.5 | 41.7 | **49.5** |
| QQP | 39.8 | 42.6 | 44.5 | 41.2 | 36.3 | **52.4** | 44.1 |
| RTE | 32.0 | 32.0 | 37.2 | 48.9 | 33.3 | **51.9** | 42.0 |
| MNLI | 36.3 | 40.6 | 41.3 | **52.5** | 43.0 | 51.0 | 48.5 |
| AG | 41.1 | 42.5 | 42.3 | 43.3 | 41.4 | 43.7 | **47.8** |
| IMDB | 45.2 | 44.0 | 43.3 | 42.0 | 44.5 | 47.9 | **48.4** |
| SCIT | 39.0 | **50.3** | 46.2 | 44.6 | 34.0 | 47.8 | 48.8 |
| SCIC | 43.9 | 44.7 | 46.3 | 41.4 | 39.1 | **48.3** | 48.1 |
| Aver. | 39.4 | 41.7 | 43.2 | 44.0 | 38.5 | **48.1** | 47.1 |

Table 8: Results of probing experiments to measure representation collapse (higher score is better). Model is fine-tuned for task A with different methods, then a new linear head is trained for the remaining 12 tasks and the mean performance is reported. Aver. is average over different choices of task A. RP$_I$ is REPINA$_I$ , RP$_M$ is REPINA$_{MLP}$ . AG: AGNEWS-10k, SCIT: SCITAIL-10k, SCIC: SCICITE-10k, QNLI: QNLI-10k, QQP:QQP-10k, MNLI: MNLI-10k.

Note that auxiliary tasks considered here are used only to evaluate the degradation of representations. They are not available during finetuning. During fine-tuning stage, only one task dataset is available. Thus, we do not compare our methods to the rehearsal-based learning methods.

## 6.2 Diversity of Fine-tuned Representations

Probing experiments rely on the availability of extra fine-tuning tasks and, thus, are limited in the amount of information they can assess, requiring additional fine-tuning rounds. Here, we propose metrics that can reliably quantify the power of fine-tuned representations by capturing their geometric diversity. The intuition behind our metrics is the following: *if all representations lie in a small dimensional space such as a straight line or a single point, then they contain little information and are not expressive. But if representations are well spread out and span the entire representation space, then they possess high information capacity*).

We illustrate representation collapse metrics from the geometric perspective in Figure 5. The top three plots show three different distributions of data points (representations). The left distribution spans all three dimensions, indicating the highest degree of data diversity. Since data points equally lie in all dimensions, all three eigenvectors ($V(\lambda_i)$'s) will be of equal importance and all three eigenvalues ($\lambda_i$'s) will be approximately equal. In contrast, the central distribution spans two axes, leading to a smaller third eigenvalue that corresponds to the "redundant" dimension. Right distribution has all the data points collapsed along one axis, resulting in

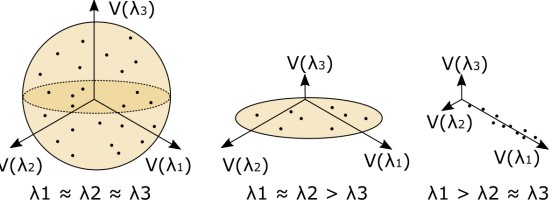

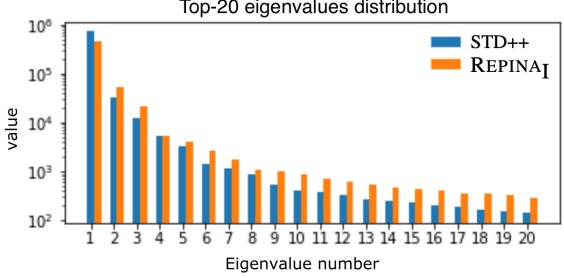

Figure 5: Top: $\lambda_i$ and $V(\lambda_i)$ correspond to $i$th eigenvalue and its associated eigenvector after eigendecomposition of Gram matrix. Data from the left distribution is well spread out and spans all three dimensions, with all of its eigenvalues being similar. The right distribution shows all of the data collapsed along one eigenvector, hence one of the eigenvalues significantly exceeds two others. Bottom: comparison of top-20 eigenvalues of STD++ and REPINA$_I$ after fine-tuning on QQP with 250 points. Less skewed distribution of eigenvalues for REPINA$_I$ compared to STD++ indicates a more spread out distribution of fine-tuned representations with RE-PINA$_I$ compared to STD++ .

one eigenvalue being substantially higher than the others. Overall, more uniform distribution of the eigenvalues corresponds to a better representation matrix diversity. In the bottom bar-plot we show distribution of the top-20 eigenvalues of the fine-tuned representations with REPINA$_I$ and STD++ after training on QQP dataset with 250 points (Figure 5). REPINA$_I$ preserved a closer to uniform eigenvalue distribution, while STD++ results in representations with much higher first eigenvalue, indicating representation collapse. Thus, REPINA$_I$ *yields better representation matrix diversity and less representation collapse than STD++* .

Next, we formalize this intuition by defining a representation diversity metric based on the geometric diversity of sentence-level representations.

**Diversity Metrics:** We compute the gram matrix $G$ for the representations where $G_{i,j} = \langle f_{fin}(x_i), f_{fin}(x_j) \rangle$. From $G$ we obtain eigenvalues $\lambda_1 \geq \ldots \geq \lambda_d$. To measure diversity of representations, we use geometric mean (GM) and

harmonic mean (HM) of the eigenvalues:

$$\text{GM} = \left(\Pi_{i=1}^{d}\lambda_i\right)^{1/d} = \text{Determinant}^{1/d}(G),$$

$$\text{HM} = \left(\sum_{i=1}^{d}\frac{1}{\lambda_i}\right)^{-1} = \text{Trace}\left(G^{-1}\right)^{-1}$$

These metrics attain a high value if the representations are well spread out and are zero or close to zero if all/most of the representations lie in a smaller dimension subspace. In contrast to arithmetic mean, geometric and harmonic mean are not as sensitive to outliers. We observe that these metrics turn out to be always zero as representations typically lie in 20-dimensional space. Hence, we chose top-$k$ $\lambda_i$ values for $k = 5, 10, 20$ where GM and HM are bounded away from 0.

$$\text{GM-k} = \left(\Pi_{i=1}^{k}\lambda_i\right)^{\frac{1}{k}}, \text{HM-k} = \left(\sum_{i=1}^{k}\frac{1}{\lambda_i}\right)^{-1}$$

We compare REPINA$_\text{I}$ and REPINA$_\text{MLP}$ to the

| Metric $\downarrow$ | STD++ | DA | WC | ReInit | R3F | RP$_\text{I}$ | RP$_\text{M}$ |
|---|---|---|---|---|---|---|---|
| GM-5 | 396 | 481 | 484 | 425 | 397 | **584** | 463 |
| GM-10 | 92 | 118 | 118 | 90 | 93 | **134** | 91 |
| GM-20 | 14 | 18 | 20 | 13 | 13 | **22** | 13 |
| HM-5 | 198 | 253 | 242 | 184 | 207 | **290** | 217 |
| HM-10 | 38 | 53 | 47 | 37 | 38 | **55** | 32 |
| HM-20 | 3 | 4 | 5 | 3 | 3 | **6** | 3 |

Table 9: Diversity of fine-tuned representations. Mean value across all the 13 tasks is presented. RP$_\text{I}$ is REPINA$_\text{I}$ , RP$_\text{M}$ is REPINA$_\text{MLP}$ . REPINA$_\text{I}$ yields fine-tuned representations with maximum representation matrix diversity.

existing baselines using GM-k and HM-k with $k = 5, 10, 20$ (Table 9). Low GM-k and HM-k indicates representation collapse, when fine-tuned representations lie in a low-dimensional space. High GM-k and HM-k indicates that representations are well spread out and span a higher dimensional space. Table 9 shows that **REPINA$_\text{I}$ results in the most diverse representations among all the baseline methods and incurs least representation collapse**(see Appendix N for detailed results).

# 7    Conclusion

In this paper, we propose a novel representation invariance regularizer targeted at avoiding representation degradation during finetuning. It has a knob that can control strength of regularization. We experiment with two choices of this knob, REPINA$_\text{I}$ and REPINA$_\text{MLP}$ and show that they both achieve significant performance gain across 13

tasks, including few-shot and label noise settings, and improve generalization performance. We also study the degradation of representations during fine-tuning, *representation collapse*, and propose new metrics to quantify it. Our methods reduce representation collapse and improve OOD robustness.

# 8    Limitations

We conduct extensive experiments in our paper and show that the proposed approaches lead to significant gains. However, we did not exhaust all avenues of investigation due to limited resources. Firstly, we could experiment with different choices of $\phi$ other than in REPINA$_\text{I}$ ($\phi$ is identity) and REPINA$_\text{MLP}$ ($\phi$ is MLP). Other choices of $\phi$ may include deeper networks or transformer-based models, which could potentially improve performance even further. Secondly, we investigated how representations from intermediate layers affect REPINA$_\text{I}$ performance, and observe major improvements with top layer representations. Similar experiments for REPINA$_\text{MLP}$ may also yield further gain. Also, in REPINA$_\text{I}$ we could experiment with more choices of the representations layer (we tried 5th, 10th, 20th layer). Since lower layer representations are more computationally efficient to regularize (do not require full forward pass through the model), another interesting direction is finding a trade-off between computational efficiency of the regularizer and performance gains.

This study primarily focused on medium-sized models due to computational constraints. It is the outcome of extensive experimentation, which would have been impractical with limited computational resources. Although we have experimented with masked language models, we believe the findings apply to other architectures that follow similar principles. We anticipate that future research will provide more insight into these issues.

## Ethical Considerations

REPINA aims to improve performance and retain the quality of representations during fine-tuning. Practically, our method can help in suppressing potential biases of language models after fine-tuning on datasets that include biased content. REPINA can achieve this by reducing collapse of representations and preserving their pre-trained knowledge. All our experiments are based on publicly available datasets and, hence, there is no immediate concern about harmful content.

## Acknowledgments

We thank the anonymous reviewers for their constructive feedback. DK is supported by generous gifts from Johns Hopkins University, Allen Institute for AI and Amazon. AR is supported by Vector Institute for Artificial Intelligence and the University of Toronto.

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

# Supplementary Material

## A    Additional Related works

Due to limited space in the main text, part of the related is below. We will reintroduce these to the main text upon having more space.

**Domain shift between pre-training and finetuning data:** Even though pretrained models achieve high performance for a large number of NLP tasks, they tend to suffer if there is a significant domain shift between the pretraining data and finetuning data. Domain Adaptation bridges this gap by adapting the model to the finetuning task domain. It can done by doing additional pre-training on task domain data if such data is available (Gururangan et al., 2020) or algorithmically finding such data from general domain corpus if such a data is not available (Madan et al., 2021).

**Domain shift between finetuning train data and evaluation data:** Domain Adaptation typically refers to the scenario where labeled train data is available in one domain and the evaluation is done for data in other domain. Techniques for addressing domain shift include model-centric techniques, data-centric techniques and hybrid techniques. Model-centric technique changes the model by changing the feature space, loss function or the structure of the model (Blitzer et al., 2006; Pan et al., 2010; Ganin et al., 2016; Ben-David et al., 2020). Data-centeric approaches involve pseudo-labeling (Abney, 2007), using auxiliary tasks (Phang et al., 2018), and data selection (Moore and Lewis, 2010; Wang et al., 2017). *Mixout* (Lee et al., 2019a) is a variant of Dropout regularization that replaces dropped neurons with the pre-trained model neurons, thereby mixing pretrained and fine-tuned parameters.

**Measures of representation:** Aghajanyan et al. (2020) measures the quality of finetuned representations by fitting them on auxiliary tasks. CKA (Kornblith et al., 2019) measures correspondences between representations from different network. Wu et al. (2020) study the similarity of internal representation and attention of different trained models using some new similarity measures.

Merchant et al. (2020) also studies what happens during finetuning via probing classifiers and representation similarity analysis. It argues that finetuning does not necessarily incurs catastrophic forgetting. It analyze the effect for finetuning different tasks on the changes in representation.

Rongali et al. (2020) show that rehearsal based learning can improve performance and perform better than Weight Consolidation. However, even though our method is inspired by multi-task learning and performs pseudo multi-task learning implicitly, we do not have access to any dataset additional to the single fine-tuning task. Thus, rehearsal based learning does not apply in our setting.

## B Theoretical Motivation and Connection to Pseudo Meta-learning

**Idea:** We view the pre-trained model as a multi-task learner with an infinite number of pseudo tasks $T_1, T_2, \ldots$. That is, for each $i$ there exists a linear layer that fits pre-trained representations to a pseudo task $T_i$. Our aim is to fine-tune the representations on a specific downstream task $B$ while preserving their ability to perform well on $T_1, T_2, \ldots$; namely, there must exist a linear model on the fine-tuned representations for each pseudo task $T_i$. The linear classification head for $T_i$ does not have to be the same for the pre-trained and fine-tuned representations, but their output should be close.

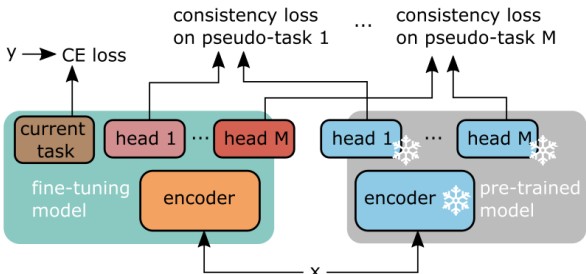

Figure 6: Intuitive explanation of the proposed approaches from the multi-task learning perspective. The total loss consists of the cross-entropy loss for the fine-tuning task and the consistency losses for the pseudo-tasks. The pre-trained model is non-trainable (frozen).

Let the training samples be $x_1, \ldots, x_N$ for the fine-tuning task $B$ and $z_{pre}^j, z_{fin}^j \in \mathbb{R}^d$ be the representations (output of the encoder layer) for the pre-trained model and the model being fine-tuned. Let $\mathcal{F}$ be a family of functions operating on representations such that each function signifies a task, $T_i$. We can formalize our objective as follows: *For any function $g \in \mathcal{F}$ on pre-trained representations, there must exists a corresponding $h \in \mathcal{F}$ on fine-tuned representations giving the same output; $\forall g \in \mathcal{F}, \exists h \in \mathcal{F}$ s.t. $g \circ z_{pre} = h \circ z_{fin}$*

During fine-tuning, we do not expect an exact agreement and allow representations to lose some expressive power. Hence, we relax the constraint and consider the representation loss error.[2]

$$\text{For } g \in \mathcal{F}, \mathcal{L}_g = \min_{h \in \mathcal{F}} \sum_{j=1}^{N} \text{loss}(g(z_{pre}^j), h(z_{fin}^j))$$

[2]If $T_i$'s were actual pre-training tasks and the data was available, we would compute the loss on the input of $T_i$. In absence of that, we approximate it by loss on (unlabeled) input of the given fine-tuning task.

If $g$ comes from a distribution $\mathcal{D}$ over $\mathcal{F}$, then our representation loss is $E_{g \sim \mathcal{D}}[\mathcal{L}_g]$. Here, we consider $\mathcal{F}$ to be the set of regression tasks[3], which are characterized by vectors in $\{u \in \mathbb{R}^d\}$ and tasks to be sampled from a standard Gaussian distribution with mean $\mathbf{0}$ and unit variance. We consider $loss$ to be the natural $\ell_2$ loss function.

$$\mathcal{L} = E_{u \sim \mathbb{N}(\mathbf{0}, \mathcal{I}_d)} \left[ \min_{v \in \mathbb{R}^d} \sum_{j=1}^{N} (u^T z_{pre}^j - v^T z_{fin}^j)^2 \right]$$

The inner minimization has a closed form solution $v = (Z_{fin}^T Z_{fin})^\dagger Z_{fin}^T Z_{pre} u$, and the resulting expectation can be reduced to get:

$$\mathcal{L} = \left\| \left( Z_{fin}(Z_{fin}^T Z_{fin})^\dagger Z_{fin}^T - I_n \right) Z_{pre} \right\|_2^2.$$

where $Z_{pre} \in \mathbb{R}^{N \times d}$ matrix has $j$-th row $z_{pre}^j$ and $Z_{fin} \in \mathbb{R}^{N \times d}$ matrix has $j$-th row $z_{fin}^j$. $||\cdot||_2$ for a vector denote the $\ell_2$-norm and for a matrix denote the Frobenius norm. $A^\dagger$ denote the pseudo-inverse of a symmetric matrix $A$. The derivation of $v$ and reduction of expectation can be found in Theorem 1 in Appendix B.1. Loss function $\hat{L}$ is not easily decomposed into mini-batches, making it challenging to optimize directly. We find an equivalent optimization problem whose objective is decomposable into mini-batches and whose optimum is equivalent to the representation loss $\mathcal{L}$:

$$\hat{\mathcal{L}} = \min_{W \in \mathbb{R}^{d \times d}} \sum_{j=1}^{N} \left\| z_{pre}^j - W z_{fin}^j \right\|_2^2$$

We can minimize $\hat{\mathcal{L}}$ for $W$ along with the fine-tuning objective. Derivation of the above equivalence can be found in Theorem 2 in Appendix B.1. We can interpret the above loss as follows: *There exists a linear function ($\phi_W : x \to Wx$) which operated on fine-tuned representations results in pre-trained representations.* We can generalize it to include a class of functions $\Phi$:

$$\hat{\mathcal{L}} = \min_{\phi \in \Phi} \sum_{j=1}^{N} \left\| z_{pre}^j - \phi(z_{fin}^j) \right\|_2^2 \qquad (3)$$

This corresponds to the aggregate loss for pseudo-tasks $T_i$'s, if instead of using a linear head for

[3]If we assume the tasks to be classification, then the linear layer is followed by a softmax layer. However, for simplicity we assume the pre-training to be done on regression tasks as this yields closed from expressions and yields good results.

pseudo tasks on fine-tuned representations, we use a function $\phi \in \Phi$ followed by $u_i$. Thus, $\Phi$ defines the strength of the regularizer. A singleton $\Phi$ containing an identity function enforces the use of the same linear head $u_i$ for task $T_i$ on both pre-trained and fine-tuned representations. This results in the strongest regularizer which keeps fine-tuned representations close to the pre-trained representations. On the other hand, if $\Phi$ is a set of very deep neural networks, then we allow a deep neural network ($+u_i$) to fit fine-tuned representations for task $T_i$. Such a neural network will almost always exist even if the fine-tuned representations have degraded significantly. Thus, it is a weak regularizer and puts mild constraints on the change of the structure of representations.

Overall, this section can be summarized as follows: (i) $\hat{\mathcal{L}}$ is an aggregate error in fitting fine-tuned representations to pseudo-pre-training tasks $T_i$'s. (ii) $\Phi$ controls the amount of structural changes in representations allowed during fine-tuning.

## B.1 Detailed Derivations

**Lemma 1.** $min_{v \in \mathbb{R}^d} \sum_{j=1}^{N}(y_j - v^T b_j)^2 = ||y - B(B^T B)^\dagger B^T y||_2^2$ where $y_j$ is the $j$-th entry of $y$.

*Proof.* Let the loss function be

$$\mathcal{L} = \sum_{j=1}^{N}(y_j - v^T b_j)^2$$

$\mathcal{L}$ is a smooth function with minimizer $v^\star$. Hence, minimum is achieved at a local minimum. Thus,

$$\frac{\delta}{\delta v}\mathcal{L}|_{v=v^\star} = \mathbf{0}$$

$$-2\sum_{j=1}^{N} b_j(y_j - (v^\star)^T b_j) = \mathbf{0}$$

$$-2\sum_{j=1}^{N} b_j y_j - b_j b_j^T v^\star = \mathbf{0}$$

$$\left(\sum_{j=1}^{N} b_j\right) y_j = \left(\sum_{j=1}^{N} b_j b_j^T\right) v^\star$$

$$\left(\sum_{j=1}^{N} b_j b_j^T\right)^\dagger \left(\sum_{j=1}^{N} b_j\right) y_j = v^\star$$

where $X^\star$ is the pseudo inverse which is equal to the inverse if $X$ is invertible. Else it spans only the space spanned by $X$. Note that $\sum_{j=1}^{N} b_j b_j^T =$

$B^T B$ and $\sum_{j=1}^{N} b_j y_j = B^T y$. So, $v^\star = (B^T B)^\dagger B^T y$. Least square error can be written in terms of vector form to get

$$min_{v \in \mathbb{R}^d} \sum_{j=1}^{N}(y_j - v^T b_j)^2 = \min_{v \in \mathbb{R}^d} ||y - Bv||_2^2$$

where $||\cdot||_2^2$ for a vector denote the $\ell_2$ norm squared. Substituting $v^*$ we get $min_{v \in \mathbb{R}^d} \sum_{j=1}^{N}(y_j - v^T b_j)^2 = ||y - B(B^T B)^\dagger B^T y||_2^2$ □

**Theorem 1.** We show that $E_{u \sim \mathbb{N}(\mathbf{0},\mathcal{I}_d)}\left[\min_{v \in \mathbb{R}^d} \sum_{j=1}^{N}(u^T z_{pre}^j - v^T z_{fin}^j)^2\right]$ $= \left|\left|\left(Z_{fin}(Z_{fin}^T Z_{fin})^\dagger Z_{fin}^T - I_n\right) Z_{pre}\right|\right|_2^2$.

*Proof.* To simplify notation, we use $a_j = z_{pre}^j$, $b_j = z_{fin}^j$, $B = Z_{fin} \in \mathbb{R}^{N \times d}$ matrix has $j$-th row $b_j$, $A = Z_{pre} \in \mathbb{R}^{N \times d}$ matrix has $j$-th row $a_j$ and $X^\dagger$ is the pseudo-inverse of $X$.

Let

$$W = E_{u \sim \mathbb{N}(\mathbf{0},\mathcal{I}_d)}\left[\min_{v \in \mathbb{R}^d} \sum_{j=1}^{N}(u^T z_{pre}^j - v^T z_{fin}^j)^2\right]$$

From Lemma 1, we get

$$W = E_{u \sim \mathbb{N}(\mathbf{0},\mathcal{I}_d)} \left|\left|Au - B(B^T B)^\dagger B^T Au\right|\right|_2^2$$

$$= E_{u \sim \mathbb{N}(\mathbf{0},\mathcal{I}_d)} \left|\left|(A - B(B^T B)^\dagger B^T A)u\right|\right|_2^2$$

**Lemma 2.** For any matrix $M$, $\mathbb{R}^{d \times d}$. $E_{u \sim \mathbb{N}(\mathbf{0},\mathcal{I}_d)}[||Mu||_2^2] = ||M||_2^2$ where $||M||_2^2$ is the forbenius norm of the matrix $M$.

*Proof.* Let the $i,j$-th entry of $M$ be $m_{i,j}$ and the $j$-th entry in $u$ be $u_j$. Then, $||Mu||_2^2 = \sum_{i=1}^{d}(\sum_{j=1}^{d} m_{i,j}u_j)^2 = \sum_{i=1}^{d} \sum_{j=1}^{d} \sum_{k=1}^{d} m_{i,j}m_{i,k}u_j u_k$.

$$E\left[||Mu||_2^2\right] = \sum_{i=1}^{d} \sum_{j=1}^{d} \sum_{k=1}^{d} m_{i,j}m_{i,k}E[u_j u_k]$$

Since $u$ is a gaussian random variable with mean 0 and covariance matrix $I_d$, we have $E[u_j u_k] = 0$ for $j \neq k$ and $E[u_i^2] = 1$ for all $i \in [d]$. Thus,

$$E\left[||Mu||_2^2\right] = \sum_{i=1}^{d} \sum_{j=1}^{d} m_{i,j}^2 = ||M||_2^2$$

□

Substituting equality from Lemma 2 to $W$, we get

$$W = \left\lVert A - B(B^TB)^\dagger B^T A \right\rVert_2^2$$

Using $\lVert -M \rVert_2^2 = \lVert M \rVert_2^2$ and substituting back $A = Z_{pre}$ and $B = Z_{fin}$, we get $E_{u \sim \mathbb{N}(\mathbf{0}, \mathcal{I}_d)} \left[ \min_{v \in \mathbb{R}^d} \sum_{j=1}^N (u^T z_{pre}^j - v^T z_{fin}^j)^2 \right]$ $= \left\lVert \left( Z_{fin}(Z_{fin}^T Z_{fin})^\dagger Z_{fin}^T - I_n \right) Z_{pre} \right\rVert_2^2$.  □

**Theorem 2.**

$$\left\lVert \left( Z_{fin}(Z_{fin}^T Z_{fin})^\dagger Z_{fin}^T - I_n \right) Z_{pre} \right\rVert_2^2$$

$$= \min_{W \in \mathbb{R}^{d \times d}} \sum_{j=1}^N \left\lVert z_{pre}^j - W z_{fin}^j \right\rVert_2^2$$

*Proof.* To simplify notation, we use $a_j = z_{pre}^j, b_j = z_{fin}^j, B = Z_{fin} \in \mathbb{R}^{N \times d}$ matrix has $j$-th row $b_j$, $A = Z_{pre} \in \mathbb{R}^{N \times d}$ matrix has $j$-th row $a_j$ and $X^\dagger$ is the pseudo-inverse of $X$. Let $W = \mathbb{R}^{d \times d}$ have $i$-th row $w_i$. We need to compute

$$L = \min_{W \in \mathbb{R}^{d \times d}} \sum_{j=1}^N \lVert a_j - W b_j \rVert_2^2$$

$$= \min_{w_1, \ldots, w_d \in \mathbb{R}^d} \sum_{j=1}^N \sum_{i=1}^d (a_{j,i} - w_i^T b_j)^2$$

$$= \sum_{i=1}^d \min_{w_i \in \mathbb{R}^d} \sum_{j=1}^N (a_{j,i} - w_i^T b_j)^2$$

where $a_{j,i}$ is the $i$-th entry of $a_j$. Applying Lemma 1, we get

$$L = \sum_{i=1}^d \left\lVert (I_n - B(B^TB)^\dagger B^T)c_i \right\rVert^2 \quad (4)$$

where $c_i$ is the $i$-th column of $A$ ($j$-th entry of $c_i$ is $a_{j,i}$).

**Lemma 3.** *For a matrix $M \in \mathbb{R}^{N \times N}$ and a set of vectors $v_1, \ldots, v_k \in \mathbb{R}^N$,*

$$\sum_{i=1}^k \lVert M v_i \rVert^2 = \lVert MV \rVert_2^2$$

*where $V \in \mathbb{R}^{N \times k}$ is the matrix with columns $v_1, \ldots, v_k$.*

*Proof.* Let $j$-th row of $M$ be $m_j$. Then,

$$\sum_{i=1}^k \lVert M v_i \rVert^2 = \sum_{i=1}^k \sum_{j=1}^N (m_j^T v)^2$$

For $j \in [N], i \in [k]$, $(j,i)$-the entry of $MV$ is $m_j^T v_i$. Thus,

$$\lVert MV \rVert^2 = \sum_{j=1}^N \sum_{i=1}^k (m_j^T v_i)^2$$

Combining the two equalities, we get

$$\sum_{i=1}^k \lVert M v_i \rVert^2 = \lVert MV \rVert_2^2$$

□

Applying Lemma 3 in eq 4, we get

$$L = \left\lVert (I_n - B(B^TB)^\dagger B^T)A \right\rVert^2$$

This finishes the proof of theorem.  □

## C  Experiment Set up details

Our implementation is based on the HuggingFace library.

### C.1  Experimental Setup

To avoid the excessively high computational cost of fine-tuning on large-scale datasets, we limited their full training sets to $10,000$ data points (marked with a suffix -10k in Table 3). For few-sample experiments, we fixed the same data subset across all models to avoid performance changes related to data variability. Since test set labels are unavailable, we use development set to report the performance.

**Batch Size:** Different methods have different memory requirement. For instance, R3F has the highest footprint which limits the batch size as we can not process too many inputs at the same time. Table 10 shows the batch size used for each dataset in our experiments.

| Task | Batch size | Task | Batch size |
|------|-----------|------|-----------|
| COLA | 4 | CHEMPROT | 1 |
| RTE | 1 | SCICITE | 2 |
| SST | 4 | SCITAIL-10k | 1 |
| MNLI-10k | 1 | AGNEWS-10k | 2 |
| MRPC | 4 | YELP-10k | 1 |
| QQP-10k | 1 | IMDB-10k | 1 |
| QNLI-10k | 1 | | |

Table 10: Batch size used in our experiments

### C.2  Filtering Failed Runs

For most of the datasets experimented here, available test data split is unlabeled. Thus, we use the validation data split to report performance. It has been observed that different finetuning runs can result in very different finetuned model performance (Mosbach et al., 2020). Thus, reporting max test run performance does not truly reflect the effectiveness of the finetuning process and the maximum test run performance across different random seeds can be substantially larger than the mean. So, in our experiments we do not use the val data (on which we report performance) to select the run or any hyperparameter. To select optimal hyperparameters such as regularization coefficient etc., we use a subset of original train data split which is not used for training. Such a data is available as we are typically finetuning with a subset of original train data. Table 11 shows the threshold for failed run for each task.

| Task | Threshold | Task | Threshold |
|------|-----------|------|-----------|
| COLA | 0.00 | CHEMPROT | 34.45 |
| RTE | 53.70 | SCICITE | 24.66 |
| SST | 54.00 | SCITAIL-10k | 60.38 |
| MNLI-10k | 30.00 | AGNEWS-10k | 10.44 |
| MRPC | 81.22 | YELP-10k | 52.80 |
| QQP-10k | 0.00 | IMDB-10k | 33.94 |
| QNLI-10k | 50.53 | | |

Table 11: Failed run threshold

## D   REPINA$_{\text{MLP}}$ - Effect of MLP depth and missing details

**Missing Details:** We use tanh activation in MLP with learning rate same as the rest of the network. Parameters of MLP are optimized alongside the language model. We use Glorot uniform initializer to initialize the parameters of MLP (Glorot and Bengio, 2010). Bias parameters are initialized to zero.

Table 12 shows that REPINA$_{\text{MLP}}$ is resistant to the number of MLP layers chosen. When training with all datapoints, performance is typically within a percentage of each other.

| Task ↓ / Num. layers → | 1 | 2 |
|---|---|---|
| IMDB-10k | 93.43 | **93.87** |
| MRPC | **91.19** | 91.12 |
| SCICITE | 82.55 | **83.15** |
| COLA | 61.86 | **62.47** |
| SST | 93.03 | **93.19** |
| MNLI-10k | 65.18 | **65.22** |
| AGNEWS-10k | 91.8 | **91.93** |
| CHEMPROT | 82.48 | **83.67** |
| QNLI-10k | 87.52 | **87.61** |
| SCITAIL-10k | 94.06 | **94.75** |
| YELP-10k | 95.66 | **95.96** |
| RTE | 72.92 | **74.37** |
| QQP-10k | 78.22 | **79.3** |
| Mean | 83.83 | 84.35 |

Table 12: Performance of REPINA$_{\text{MLP}}$ with different number of MLP layers.

## E   REPINA$_{\text{I}}$ - Which layer to regularize?

Table 13 compares the result between regularizing the top layer vs regularizing the intermediate layer in REPINA$_{\text{I}}$ . We observe that REPINA$_{\text{I}}$ consistently outperform when regularizing the intermediate layer.

Table 14 shows the the result for REPINA$_{\text{I}}$ with representations chosen from 5th, 10th or 20th layer of encoder. Note that 5th layer is the closest to the input and doesn't account for token embedding layer. We note that all three choices are performing roughly equally well. Mean performance is typically less than a percentage point from each other. If one were to use a single layer, one can use 5th for low-data case and 10th or 20th for large dataset case.

| Tasks | STD++ | Top | Interm. |
|---|---|---|---|
| 500 Training Datapoints | | | |
| QNLI | 80.86 | 80.7 | **82.56** |
| MNLI | 41.74 | 41.07 | **49.07** |
| AGNEWS | 89.09 | 89.34 | **89.53** |
| IMDB | 83.65 | 89.77 | **91.32** |
| SST | 89.54 | 89.49 | **89.68** |
| COLA | 45.29 | 44.22 | **46.95** |
| CHEMPROT | 65.59 | 61.0 | **73.47** |
| MRPC | 84.44 | 84.54 | **85.26** |
| SCITAIL | 85.11 | 88.97 | **90.07** |
| SCICITE | 78.84 | 79.36 | **79.45** |
| RTE | 61.01 | 59.39 | **62.45** |
| YELP | **93.32** | 89.02 | 93.2 |
| QQP | 61.12 | 69.5 | **70.97** |
| Mean | 73.81 | 74.34 | **77.23** |
| Average Rank | 2.38 | 2.54 | **1.08** |

Table 13: Performance for REPINA$_{\text{I}}$ -intermediate vs REPINA$_{\text{I}}$ -top.

| Tasks | STD++ | 5 | 10 | 20 |
|---|---|---|---|---|
| 500 Training datapoints | | | | |
| COLA | 45.29 | **46.95** | 44.58 | 44.34 |
| QNLI | 80.86 | **82.56** | 81.38 | 80.7 |
| MRPC | 84.44 | **85.26** | 84.78 | 84.31 |
| SST | 89.54 | **89.68** | 89.24 | 89.66 |
| SCITAIL | 85.11 | **90.36** | 90.07 | 89.2 |
| YELP | **93.32** | 92.88 | 93.2 | 93.2 |
| AGNEWS | 89.09 | **89.53** | 89.19 | 88.99 |
| RTE | 61.01 | 62.45 | **63.63** | 61.52 |
| MNLI | 41.74 | 44.61 | **49.07** | 44.57 |
| QQP | 61.12 | **72.6** | 71.79 | 70.97 |
| IMDB | 83.65 | 90.84 | 90.31 | **91.32** |
| CHEMPROT | 65.59 | 73.34 | **73.47** | 71.98 |
| SCICITE | 78.84 | 79.39 | 78.76 | **79.45** |
| Mean | 73.81 | 76.96 | 76.88 | 76.17 |

Table 14: Effect of embedding layer to be regularized in REPINA$_{\text{I}}$ .

## F Hyperparameter Optimization over learning rate, epochs and dropout

Table 15 shows the results when we search over optimal learning rate and number of epochs for each task and method. For learning rate, we perform the search over [5e-6,1e-5,2e-5,4e-5] and for epochs, we search over [5, 10]. We add an additional baseline method $DR$ where we search over dropout rate from $[0.05, 0.1, 0.2, 0.4]$.

| Tasks | STD++ | DR | DA | WC | ReInit | R3F | $RP_I$ | $RP_M$ |
|-------|-------|------|------|------|--------|------|--------|--------|
| QQP   | 78.6  | 78.9 | 78.8 | 78.2 | 78.7   | 78.6 | **79.7** | 79.2 |
| RTE   | 72.9  | 73.4 | 74.4 | 74.3 | 74.1   | 74.2 | **75.1** | 75.0 |
| COLA  | 56.7  | 61.2 | 62.4 | 61.8 | 61.5   | 61.7 | **62.5** | 61.8 |
| MRPC  | 90.3  | 90.8 | 90.5 | 90.4 | 90.8   | 90.4 | 90.8   | **90.8** |

Table 15: Results with HPO over epochs and learning rate. DR is a baseline method where we do additional HPO over dropout rate as well.

## G Experiments for RoBERTa

In the results above, we observed that our methods improve significant gain over baseline methods for BERT-large. Table 16 shows the result when we compare REPINA$_I$ against STD++ . We finetune the model for 10 epochs with regularization coefficient of 0.01 and learning rate 1e-5. Mean and standard deviation across three runs is reported. We observe that REPINA$_I$ improved STD++ performance in all cases.

| Tasks | STD++ | $RP_I$ |
|-------|-------|--------|
| MRPC  | $90.3 \pm 1.0$ | $92.0 \pm 0.5$ |
| RTE   | $74.1 \pm 2.1$ | $77.1 \pm 0.7$ |
| CoLA  | $60.0 \pm 1.1$ | $60.1 \pm 0.6$ |

Table 16: Results for RoBERTa-base on 3 GLUE datasets.

## H  Supervised Contrastive Learning

Let a mini-batch has $m$ examples, $(x_1, y_1), \ldots, (x_m, y_m)$ and $z_1, \ldots, z_m$ be the representations (output of encoder) using the model being finetuned. Supervised Contrastive Learning encourages the representations of examples of same label in the mini-batch to be close to each other and far from the examples with different label by adding the following loss to the objective:

$$\mathcal{L}_{SCL} = \sum_{i=1}^{m} -\frac{1}{N_{y_i} - 1} \sum_{j=1}^{m} 1_{i \neq j} 1_{y_i = y_j}$$
$$\log \frac{exp(\langle z_i, z_j \rangle / \tau)}{\sum_{k=1}^{m} 1_{i \neq k} exp(\langle z_i, z_k \rangle / \tau)}$$

where $\tau$ is a scalable temperature parameter that controls the separation of classes. Loss function during training is

$$\mathcal{L} = \lambda \mathcal{L}_{CE} + (1 - \lambda) \mathcal{L}_{SCL}$$

where $\mathcal{L}_{CE}$ is the cross entropy loss where $\lambda$ is a factor that can be tuned. This was shown to improve finetuning process in (Gunel et al., 2020) for few-shot finetuning.

From the definition of $\mathcal{L}_{SCL}$ we observe that SCL is only effective when the mini-batch size is large and each label class is sufficiently represented in the mini-batch. Otherwise, the loss function $\mathcal{L}_{SCL}$ is vacuous. For instance, if the mini-batch size is 1 which is the case for many of our datasets, then $\hat{L}_{SCL} = 0$ for all the mini-batches. Thus, it is equivalent to the standard finetuning. Large mini-batch size however requires large memory during finetuning process which is not always available as in our case. Thus, we look for a relaxation of SCL which can be implemented in a memory efficient manner.

We start by considering $\mathcal{L}_{SCL}$ over the entire input set instead of mini-batch and then replace the example $x_j$ with mean of examples of the same class as $x_j$ while computing similarity with another example. More formally, let the training data be $(x_1, y_1), \ldots, (x_N, y_n)$, the set of labels be $\{1, \ldots, \ell\}$ and representation of $x_i$ from the encoder of finetuning model. Let $C_j = \{i \mid y_i = j\}$ and $c_j = \frac{1}{|C_j|} \sum_{i \in C_j} z_i$ be the center of embeddings of inputs with label $j$. We consider the fol-

| Tasks | STD++ | SCL | REPINA$_I$ | REPINA$_{MLP}$ |
|---|---|---|---|---|
| | 250 datapoints | | | |
| QNLI | 75.11 | 73.79 | **78.82** | 76.13 |
| SST | 88.41 | 87.27 | **89.29** | 88.59 |
| QQP | 68.28 | 68.79 | 68.76 | **70.38** |
| SCITAIL | 82.31 | 86.13 | **88.9** | 86.5 |
| MNLI | 37.7 | 38.07 | 38.35 | **41.18** |
| IMDB | 86.11 | 90.27 | **90.38** | 90.33 |
| RTE | 59.13 | 60.77 | **60.83** | 59.3 |
| MRPC | 84.43 | **85.67** | 84.65 | 84.08 |
| COLA | 41.57 | 31.91 | 43.98 | **45.28** |
| CHEMPROT | 55.22 | 32.0 | **63.28** | 62.32 |
| Mean | 67.83 | 65.47 | **70.72** | 70.41 |
| Average Rank | 4.3 | 3.7 | **1.6** | 2.35 |

Table 17: Performance of memory-efficiet SCL.

lowing relaxation of $\mathcal{L}_{SCL}$.

$$\hat{\mathcal{L}}_{SCL} = \sum_{i=1}^{N} -\frac{1}{N_{y_i} - 1} \sum_{j=1}^{N} 1_{i \neq j} 1_{y_i = y_j}$$
$$\log \frac{exp(\langle z_i, c_{y_j} \rangle / \tau)}{\sum_{k=1}^{N} 1_{i \neq k} exp(\langle z_i, c_{y_k} \rangle / \tau)}$$
$$= -\sum_{i=1}^{N} \sum_{j=1}^{\ell} 1_{j \neq y_i}$$
$$\log \frac{exp(\langle z_i, c_j \rangle / \tau}{\sum_{k=1}^{\ell} 1_{k \neq y_i} |C_k| exp(\langle z_i, c_k \rangle / \tau)}$$

A naive implementation of this loss function would be very expansive as the centers $c_1, \ldots, c_\ell$ would change in each iteration. We observe that centers change much slower than the individual examples. This is the reason to replace individual training samples with the centers while computing similarity $\langle z_i, z_j \rangle$. Thus, we do not update it in each iteration and instead update it only ten times during the finetuning process. Note that it increases the training time by roughly a factor of 10 which is also prohibitive for large datasets. Table 17 shows the comparison of memory efficient SCL with our methods. We see that both REPINA$_I$ and REPINA$_{MLP}$ beat SCL consistently . Moreover, SCL incur significant loss for several datasets.

# I  Comparison of REPINA$_I$ and REPINA$_{MLP}$ against each baseline

Table 18 show that both REPINA$_I$ and RE-PINA$_{MLP}$ outperform each baseline method in majority of the datasets/

| | REPINA$_I$ | REPINA$_{MLP}$ |
|---|---|---|
| **# wins against baselines methods** | | |
| **GLUE datasets (out of 7)** | | |
| STD++ | 7 | 6 |
| DA | 5 | 5 |
| WC | 5 | 6 |
| ReInit | 7 | 7 |
| R3F | 7 | 7 |
| **Non-GLUE datasets (out of 6)** | | |
| STD++ | 6 | 6 |
| DA | 6 | 6 |
| WC | 3 | 5 |
| ReInit | 6 | 6 |
| R3F | 6 | 6 |

Table 18: Number of tasks for which REPINA$_I$ or RE-PINA$_{MLP}$ outperform the baseline method.

# J  Small dataset results

Table 19 and 20 show the performance for few-sample finetuning setting.

| Tasks | STD++ | DA | WC | ReInit | R3F | REPINA$_I$ | REPINA$_{MLP}$ |
|---|---|---|---|---|---|---|---|
| | | | 250 Training datapoints | | | | |
| QQP | 68.28 | 65.63 | 69.01 | 68.46 | 67.45 | 68.76 | **70.38** |
| COLA | 41.57 | 40.39 | 45.14 | 43.73 | 40.76 | 43.98 | **45.28** |
| RTE | 59.13 | **61.37** | 58.99 | 60.29 | 60.02 | 60.83 | 59.3 |
| MNLI | 37.7 | 40.24 | 33.71 | **41.47** | 37.8 | 38.35 | 41.18 |
| YELP | 92.51 | 92.68 | 92.97 | **93.64** | 93.13 | 93.13 | 93.3 |
| CHEMPROT | 55.22 | 58.6 | 59.52 | **64.92** | 59.6 | 63.28 | 62.32 |
| QNLI | 75.11 | 78.72 | 75.27 | 77.62 | 77.43 | **78.82** | 76.13 |
| IMDB | 86.11 | 89.17 | 89.51 | 89.44 | 89.09 | **90.38** | 90.33 |
| SCICITE | 76.86 | 77.2 | 76.23 | **78.86** | 77.39 | 76.11 | 75.76 |
| SST | 88.41 | 88.1 | 88.14 | 88.3 | 88.47 | **89.29** | 88.59 |
| MRPC | 84.43 | 83.81 | **84.93** | 84.87 | 84.71 | 84.65 | 84.08 |
| SCITAIL | 82.31 | 87.65 | 86.2 | 87.8 | 88.6 | **88.9** | 86.5 |
| AGNEWS | 88.08 | 87.51 | 87.55 | 87.36 | 87.99 | **88.53** | 88.13 |
| Mean | 71.98 | 73.16 | 72.86 | **74.37** | 73.27 | 74.23 | 73.94 |
| Average Rank | 5.62 | 4.92 | 4.62 | 3.0 | 4.04 | **2.5** | 3.31 |

Table 19: Performance for different regularization methods.

| Tasks | STD++ | DA | WC | ReInit | R3F | REPINA$_I$ | REPINA$_{MLP}$ |
|---|---|---|---|---|---|---|---|
| | | | 500 Training datapoints | | | | |
| QQP | 61.12 | 71.5 | 70.13 | **72.03** | 71.61 | 70.97 | 71.67 |
| COLA | 45.29 | 46.27 | **47.34** | 46.03 | 45.36 | 46.95 | 46.76 |
| RTE | 61.01 | 60.53 | 61.23 | **63.33** | 60.41 | 62.45 | 62.94 |
| MNLI | 41.74 | 49.05 | 46.32 | **51.11** | 42.25 | 49.07 | 44.73 |
| YELP | 93.32 | **93.46** | 92.87 | 93.04 | 93.27 | 93.2 | 93.35 |
| CHEMPROT | 65.59 | **74.2** | 70.78 | 73.2 | 70.04 | 73.47 | 72.9 |
| QNLI | 80.86 | 82.4 | 82.19 | 81.84 | 82.42 | **82.56** | 82.43 |
| IMDB | 83.65 | 90.42 | **91.47** | 90.57 | 89.45 | 91.32 | 90.49 |
| SCICITE | 78.84 | 79.19 | 79.21 | 79.14 | 78.38 | 79.45 | **79.8** |
| SST | 89.54 | 89.36 | 90.02 | **90.37** | 90.1 | 89.68 | 89.4 |
| MRPC | 84.44 | 84.69 | 85.4 | 84.93 | 84.77 | 85.26 | **85.62** |
| SCITAIL | 85.11 | 87.41 | **90.23** | 89.38 | 80.73 | 90.07 | 89.3 |
| AGNEWS | 89.09 | 88.93 | 89.0 | 89.09 | **89.59** | 89.53 | 89.54 |
| Mean | 73.81 | 76.72 | 76.63 | **77.24** | 75.26 | 77.23 | 76.84 |
| Average Rank | 6.08 | 4.38 | 3.69 | 3.31 | 4.85 | **2.77** | 2.92 |

Table 20: Performance for different regularization methods.

# K  Results without any filtered runs

| | STD++ | DA | WC | ReInit | R3F | $RP_I$ | $RP_M$ |
|---|---|---|---|---|---|---|---|
| Successful runs (Failed runs filtered) | | | | | | | |
| Mean | 81.16 | 83.74 | 83.58 | 83.3 | 82.55 | 84.01 | **84.36** |
| std | 2.87 | 0.56 | 0.83 | 0.76 | 2.19 | 0.61 | **0.49** |
| Frac | 0.34 | 0.06 | 0.18 | 0.05 | 0.11 | **0.00** | 0.04 |
| All runs (Failed runs not filtered) | | | | | | | |
| Mean | 64.71 | 78.58 | 81.4 | 79.97 | 72.97 | **83.99** | 83.04 |
| std | 13.4 | 6.14 | 2.89 | 5.89 | 8.62 | **0.60** | 1.28 |

Table 21: Stability of fine-tuning results. $RP_I$ : RE-PINA$_I$ , $RP_M$ : REPINA$_{MLP}$ . Frac: fraction of fine-tuning runs filtered due to low performance. Mean, std: mean and standard deviation in performance across all datasets.

Table 22, and 23 shows performance without filtering out failed runs.

| Tasks | STD++ | DA | WC | ReInit | R3F | REPINA$_I$ | REPINA$_{MLP}$ |
|---|---|---|---|---|---|---|---|
| 250 datapoints | | | | | | | |
| YELP | 92.51 | 92.68 | 92.97 | **93.64** | 93.13 | 93.13 | 93.57 |
| RTE | 55.6 | 58.12 | 55.96 | 60.29 | 58.88 | 59.78 | 60.14 |
| MNLI | 25.75 | 30.22 | 28.42 | **41.47** | 31.94 | 39.2 | 38.58 |
| QNLI | 72.71 | 77.5 | 75.27 | 77.62 | 77.43 | **78.82** | 76.13 |
| SCITAIL | 82.31 | 87.65 | 86.2 | 87.8 | 88.6 | **88.9** | 86.5 |
| SST | 88.41 | 88.1 | 88.14 | 88.3 | 88.47 | **89.29** | 88.59 |
| CHEMPROT | 55.22 | 58.6 | 59.52 | **64.92** | 59.6 | 60.72 | 62.32 |
| AGNEWS | 88.08 | 87.51 | 87.55 | 87.36 | 87.99 | **88.53** | 88.13 |
| SCICITE | 76.86 | 77.2 | 76.23 | 78.86 | 77.39 | **78.95** | 75.76 |
| IMDB | 83.5 | 80.13 | 89.51 | 89.44 | 89.05 | **90.38** | 90.33 |
| COLA | 41.57 | 40.39 | 45.14 | 43.73 | 40.76 | 43.98 | **45.28** |
| MRPC | 84.43 | 83.81 | **84.93** | 84.87 | 84.71 | 84.65 | 84.08 |
| QQP | 35.94 | 64.08 | 51.92 | 68.46 | 55.81 | **68.76** | 65.5 |
| Mean | 67.91 | 71.23 | 70.9 | **74.37** | 71.83 | 74.24 | 73.45 |
| Average Rank | 5.92 | 5.46 | 4.85 | 2.69 | 3.88 | **1.96** | 3.23 |

Table 22: Performance for different regularization methods without filtering failed runs.

| Tasks | STD++ | DA | WC | ReInit | R3F | REPINA$_I$ | REPINA$_{MLP}$ |
|---|---|---|---|---|---|---|---|
| All datapoints | | | | | | | |
| MRPC | 86.84 | 90.67 | 88.6 | 90.98 | 89.9 | **91.49** | 91.12 |
| IMDB-10k | 66.59 | 93.24 | 93.69 | 92.7 | 93.1 | **93.96** | 93.87 |
| YELP-10k | 72.65 | 95.55 | 95.81 | 95.56 | 95.42 | 95.78 | **95.96** |
| SCICITE | 81.87 | 82.13 | 82.23 | 82.41 | 81.97 | 82.74 | **83.15** |
| QNLI-10k | 64.93 | 81.64 | 86.1 | 86.73 | 78.43 | 86.85 | **87.36** |
| CHEMPROT | 72.59 | 82.57 | **83.91** | 82.46 | 82.73 | 83.49 | 83.67 |
| MNLI-10k | 14.37 | 45.98 | 55.94 | 46.54 | 21.66 | **65.48** | 64.81 |
| COLA | 59.69 | **63.45** | 61.5 | 61.25 | 62.04 | 62.34 | 62.47 |
| RTE | 51.35 | 56.14 | 52.87 | 66.86 | 56.68 | **71.26** | 61.44 |
| AGNEWS-10k | 91.67 | 91.82 | 91.92 | 91.67 | 91.73 | **92.07** | 91.93 |
| SST | 81.95 | 84.13 | 92.32 | 92.28 | 83.88 | 92.71 | **93.23** |
| QQP-10k | 5.9 | 47.77 | 76.25 | 55.16 | 27.8 | 79.03 | **79.3** |
| SCITAIL-10k | 76.01 | 93.74 | 94.03 | 93.36 | 86.54 | 93.74 | **94.75** |
| Mean | 63.57 | 77.6 | 81.17 | 79.84 | 73.22 | **83.92** | 83.31 |
| Average Rank | 6.92 | 4.38 | 3.46 | 4.38 | 5.31 | 1.92 | **1.62** |

Table 23: Performance for different regularization methods without filtering failed runs.

## L    Detailed results for label noise

Table 24, 25, 26 shows detailed results with varying amount of label noise in the training data.

| Tasks | STD++ | DA | WC | ReInit | R3F | REPINA$_I$ | REPINA$_{MLP}$ |
|---|---|---|---|---|---|---|---|
| | | | Noise level = 0.05 | | | | |
| SCICITE | 81.29 | 56.46 | 81.34 | 81.11 | 81.03 | 81.36 | **81.88** |
| QNLI-10k | 57.35 | 50.18 | 68.59 | 67.16 | 64.33 | 83.87 | **86.08** |
| MRPC | 87.4 | 87.48 | 86.66 | **89.11** | 88.41 | 88.41 | 86.81 |
| RTE | 51.35 | 48.59 | 51.55 | 65.63 | 48.65 | **67.58** | 61.16 |
| IMDB-10k | 72.43 | 68.28 | 77.4 | 62.14 | 91.63 | **92.84** | 92.56 |
| CHEMPROT | 71.57 | 81.42 | **83.88** | 81.81 | 81.48 | 81.64 | 82.18 |
| AGNEWS-10k | 91.1 | 91.11 | **91.55** | 91.07 | 91.26 | 91.21 | 91.47 |
| YELP-10k | 49.92 | 72.44 | 86.24 | 85.4 | 72.38 | 95.1 | **95.36** |
| MNLI-10k | 0.0 | 0.0 | 31.9 | 41.99 | 47.51 | **63.26** | 24.96 |
| SST-10k | 91.12 | 83.17 | **91.83** | 90.05 | 91.31 | 83.21 | 90.73 |
| SCITAIL-10k | 58.21 | 57.94 | 92.91 | 83.36 | 83.93 | 92.32 | **93.48** |
| QQP-10k | 0.0 | 0.0 | 76.24 | 57.38 | 0.0 | 77.78 | **78.82** |
| COLA | 46.38 | 47.84 | 59.33 | 43.47 | 45.15 | **59.53** | 48.29 |
| Mean | 58.32 | 57.3 | 75.34 | 72.28 | 68.24 | **81.39** | 77.98 |
| Average Rank | 5.5 | 5.96 | 2.92 | 4.38 | 4.35 | **2.42** | 2.46 |

Table 24: Training with at most 10k training datapoints on 13 datasets.

| Tasks | STD++ | DA | WC | ReInit | R3F | REPINA$_I$ | REPINA$_{MLP}$ |
|---|---|---|---|---|---|---|---|
| | | | Noise level = 0.1 | | | | |
| SST-10k | 88.0 | 66.93 | **91.58** | 88.97 | 81.02 | 82.73 | 82.64 |
| MNLI-10k | 12.44 | 0.13 | 57.55 | 48.92 | 24.31 | **59.67** | 12.38 |
| SCITAIL-10k | 65.86 | 63.01 | 74.68 | 80.92 | 69.86 | 90.97 | **92.82** |
| IMDB-10k | 33.33 | 33.33 | 44.86 | 67.47 | 76.92 | 90.96 | **91.88** |
| RTE | 50.54 | 49.46 | 48.38 | 61.52 | 50.54 | **64.77** | 60.58 |
| MRPC | 84.11 | 83.69 | 85.11 | **88.49** | 84.33 | 86.06 | 87.4 |
| AGNEWS-10k | 90.19 | 90.24 | 90.54 | 90.23 | 90.32 | 90.51 | **90.58** |
| CHEMPROT | 80.75 | 68.56 | 82.25 | 80.56 | 69.38 | **82.49** | 71.92 |
| QQP-10k | 0.0 | 0.0 | 75.49 | 14.99 | 14.92 | 76.17 | **76.76** |
| YELP-10k | 60.78 | 60.77 | 83.65 | 76.4 | 73.85 | 94.22 | **94.64** |
| COLA | 56.75 | 45.85 | **58.65** | 52.8 | 45.43 | 44.12 | 55.6 |
| QNLI-10k | 50.32 | 50.54 | 65.61 | 63.15 | 66.69 | 74.24 | **83.83** |
| SCICITE | 80.79 | 69.21 | 80.45 | 80.71 | 80.29 | 78.76 | **81.52** |
| Mean | 57.99 | 52.44 | 72.22 | 68.86 | 63.68 | **78.13** | 75.58 |
| Average Rank | 4.96 | 6.46 | 3.23 | 3.46 | 4.73 | 2.77 | **2.38** |

Table 25: Training with at most 10k training datapoints on 13 datasets.

| Tasks | STD++ | DA | WC | ReInit | R3F | REPINA$_I$ | REPINA$_{MLP}$ |
|---|---|---|---|---|---|---|---|
| | | | Noise level = 0.3 | | | | |
| QQP-10k | 0.0 | 0.0 | **68.62** | 13.08 | 16.01 | 68.31 | 67.36 |
| SCICITE | 34.18 | 24.67 | 43.99 | **73.7** | 62.73 | 67.3 | **73.7** |
| COLA | 3.42 | 6.67 | **35.97** | 22.14 | 12.8 | 15.01 | 24.99 |
| IMDB-10k | 33.33 | 33.33 | **47.15** | 43.14 | 33.33 | 33.33 | 35.6 |
| MNLI-10k | 0.0 | 0.0 | **1.29** | 0.0 | 0.0 | 0.22 | 0.0 |
| CHEMPROT | 41.96 | 54.1 | 69.21 | **75.81** | 45.41 | 67.7 | 53.96 |
| QNLI-10k | 50.11 | 49.89 | 56.27 | 55.13 | 50.27 | 50.18 | **77.96** |
| AGNEWS-10k | 70.31 | **83.14** | 67.91 | 81.75 | 82.41 | 45.87 | 68.06 |
| RTE | 49.46 | 49.46 | 51.48 | 53.14 | 49.46 | **60.83** | 54.51 |
| YELP-10k | 50.02 | 49.56 | 56.02 | 60.22 | 49.79 | 51.99 | **91.44** |
| SCITAIL-10k | 49.62 | 49.62 | 58.45 | 49.62 | 49.62 | 56.83 | **86.71** |
| MRPC | 80.92 | **81.38** | 81.26 | 78.42 | 81.22 | 79.54 | 80.63 |
| SST-10k | 58.23 | 64.04 | 57.73 | **75.53** | 63.03 | 57.08 | 57.98 |
| Mean | 40.12 | 41.99 | 53.49 | 52.44 | 45.85 | 50.32 | **59.45** |
| Average Rank | 5.5 | 4.88 | **2.77** | 3.23 | 4.54 | 4.04 | 3.04 |

Table 26: Training with at most 10k training datapoints on 13 datasets.

# M Representation Collapse - Continual learning perspective

Table 27, 28, 29 shows results for representation collapse when we finetune the model for task $A$ using different methods and then finetune the top layer for task $B$.

| Tasks | STD++ | DA | WC | ReInit | R3F | REPINA$_I$ | REPINA$_{MLP}$ |
|---|---|---|---|---|---|---|---|
| COLA | -0.38 | -0.63 | 3.39 | -1.17 | 0.0 | 0.93 | **5.08** |
| MRPC | 81.62 | 81.22 | **83.13** | 81.57 | 81.44 | 82.12 | 82.42 |
| QQP-10k | 29.68 | 28.97 | 43.13 | 31.71 | 22.18 | 46.74 | **59.26** |
| YELP-10k | 50.86 | 51.55 | 55.13 | 51.46 | 50.99 | 52.63 | **60.04** |
| SCITAIL-10k | 59.55 | 58.44 | **74.46** | 61.32 | 49.62 | 67.38 | 71.24 |
| SCICITE | 24.82 | 24.84 | **28.74** | 24.75 | 24.67 | 25.02 | 27.71 |
| AGNEWS-10k | 20.22 | 20.5 | 36.61 | 16.92 | 20.15 | 31.01 | **62.9** |
| IMDB-10k | 41.36 | 33.33 | 49.87 | 41.0 | 43.36 | 43.57 | **55.72** |
| CHEMPROT | 33.1 | 32.46 | **33.23** | 32.92 | 32.34 | 33.09 | 33.15 |
| MNLI-10k | 8.07 | 8.27 | 16.76 | 8.36 | 6.92 | 14.58 | **17.69** |
| SST-10k | 51.25 | 54.55 | 58.21 | 52.24 | 52.2 | 55.93 | **66.36** |
| RTE | 51.32 | 51.62 | 51.5 | 51.26 | **53.55** | 47.83 | 52.17 |
| Mean | 37.62 | 37.09 | 44.51 | 37.69 | 36.45 | 41.74 | **49.48** |
| Average Rank | 5.17 | 5.17 | 1.92 | 5.33 | 5.67 | 3.33 | **1.42** |

Table 27: Results for training top layer for different task after finetuning entire model for QNLI-10k

| Tasks | STD++ | DA | WC | ReInit | R3F | REPINA$_I$ | REPINA$_{MLP}$ |
|---|---|---|---|---|---|---|---|
| MRPC | 81.45 | 81.4 | **82.67** | 81.18 | 81.22 | 82.13 | 81.15 |
| CHEMPROT | 33.46 | 33.46 | 33.46 | 33.46 | 33.46 | **33.59** | 33.46 |
| QNLI-10k | 59.06 | 63.74 | **67.98** | 61.65 | 50.54 | 64.95 | 64.58 |
| YELP-10k | 50.18 | 50.02 | 55.91 | 51.43 | 50.06 | **64.07** | 51.32 |
| SCITAIL-10k | 62.54 | 72.01 | 68.1 | 65.55 | 49.62 | **76.55** | 71.55 |
| COLA | 0.23 | -0.79 | 0.0 | -0.01 | 0.0 | **5.17** | -0.82 |
| AGNEWS-10k | 16.02 | 19.77 | 32.56 | 19.6 | 10.76 | **64.32** | 38.5 |
| IMDB-10k | 39.92 | 45.49 | 45.45 | 39.17 | 33.33 | **66.28** | 34.55 |
| SCICITE | 24.67 | 24.67 | 24.65 | 24.67 | 24.67 | **28.51** | 24.67 |
| MNLI-10k | 11.73 | 17.18 | 18.91 | 15.72 | 0.0 | **19.57** | 19.38 |
| SST-10k | 49.69 | nan | 52.24 | 50.89 | 51.03 | **69.87** | 58.08 |
| RTE | 48.86 | 52.71 | 52.89 | 51.44 | 50.9 | **53.52** | 52.17 |
| Mean | 39.82 | 41.79 | 44.57 | 41.23 | 36.3 | **52.38** | 44.05 |
| Average Rank | 4.96 | 3.79 | 3.08 | 4.79 | 5.67 | **1.17** | 4.04 |

Table 28: Results for training top layer for different task after finetuning entire model for QQP-10k

| Tasks | STD++ | DA | WC | ReInit | R3F | REPINA$_I$ | REPINA$_{MLP}$ |
|---|---|---|---|---|---|---|---|
| COLA | 0.0 | 0.0 | 1.48 | -0.59 | 0.0 | **5.06** | 1.91 |
| MRPC | 81.22 | 81.22 | 81.31 | 80.75 | 81.22 | 81.68 | **81.91** |
| QQP-10k | 0.0 | 0.0 | 18.89 | 60.51 | 0.0 | **60.87** | 36.29 |
| QNLI-10k | 50.54 | 50.54 | 57.24 | 58.24 | 50.54 | **62.8** | 55.56 |
| YELP-10k | 50.02 | 50.02 | 50.15 | 58.96 | 53.13 | **63.63** | 57.1 |
| SCITAIL-10k | 49.62 | 49.62 | 57.21 | 79.02 | 49.62 | **82.36** | 67.64 |
| AGNEWS-10k | 10.0 | 10.0 | 21.94 | 44.49 | 15.14 | **52.33** | 21.44 |
| IMDB-10k | 33.33 | 33.33 | 38.1 | 56.09 | 35.22 | **57.18** | 44.89 |
| SCICITE | 24.67 | 24.67 | 25.14 | 25.33 | 24.67 | **31.81** | 25.96 |
| CHEMPROT | 33.46 | 33.46 | 33.46 | 33.26 | 33.46 | **33.56** | 33.46 |
| MNLI-10k | 0.0 | 0.0 | 6.4 | **29.24** | 2.67 | 27.0 | 20.51 |
| SST-10k | 50.92 | 50.92 | 55.59 | 60.92 | 53.36 | **63.88** | 56.86 |
| Mean | 31.98 | 31.98 | 37.24 | 48.85 | 33.25 | **51.85** | 41.96 |
| Average Rank | 5.96 | 5.88 | 3.75 | 3.25 | 5.08 | **1.17** | 2.92 |

Table 29: Results for training top layer for different task after finetuning entire model for RTE

# N Measuring representation collapse

Table 30 show the sum of top-k normalized eigen-values (divide each eigenvalue by the sum of eigen-values) for k=10. From this, we can observe that almost all the normalized eigenvalues after the first twenty are close to zero

| Tasks | STD++ | DA | WC | ReInit | R3F | REPINA$_I$ | REPINA$_{MLP}$ |
|---|---|---|---|---|---|---|---|
| RTE | **1.0** | 1.0 | 1.0 | 0.99 | 1.0 | 0.99 | 1.0 |
| MRPC | 1.0 | 0.99 | 0.99 | 0.96 | 1.0 | 1.0 | **1.0** |
| QNLI-10k | 1.0 | 1.0 | 1.0 | **1.0** | 1.0 | 1.0 | 1.0 |
| SCITAIL-10k | **1.0** | 1.0 | 1.0 | 1.0 | 1.0 | 1.0 | 1.0 |
| IMDB-10k | 1.0 | 1.0 | 1.0 | **1.0** | 1.0 | 1.0 | 1.0 |
| SST-10k | 1.0 | 1.0 | 0.99 | **1.0** | 1.0 | 1.0 | 1.0 |
| COLA | 0.99 | 0.99 | 0.98 | 0.99 | **1.0** | 1.0 | 1.0 |
| AGNEWS-10k | 0.98 | 0.98 | 0.98 | **0.99** | 0.98 | 0.98 | 0.99 |
| QQP-10k | 1.0 | 1.0 | 1.0 | 1.0 | **1.0** | 0.99 | 1.0 |
| MNLI-10k | **1.0** | 0.99 | 0.99 | 0.99 | 1.0 | 0.99 | 1.0 |
| YELP-10k | 1.0 | 1.0 | 1.0 | **1.0** | 1.0 | 1.0 | 1.0 |
| CHEMPROT | 0.98 | 0.97 | 0.97 | 0.98 | 0.98 | 0.98 | **0.99** |
| SCICITE | 0.99 | 0.99 | 0.98 | **1.0** | 1.0 | 0.98 | 0.99 |

Table 30: Normalized average of top-10 eigenvalues

Tables 31 and 32 show the GM-k and HM-k for k=10. We observe that REPINA$_I$ achieves the highest value and thus is most effective in reducing representation collapse.

| Tasks | STD++ | DA | WC | ReInit | R3F | REPINA$_I$ | REPINA$_{MLP}$ |
|---|---|---|---|---|---|---|---|
| RTE | 0.01 | 0.01 | 28.58 | 54.68 | 0.54 | **113.27** | 24.64 |
| MRPC | 47.83 | 85.21 | **108.67** | 65.41 | 59.4 | 61.13 | 19.25 |
| QNLI-10k | 29.3 | 34.28 | **54.86** | 16.92 | 29.65 | 47.15 | 46.11 |
| SCITAIL-10k | 25.23 | **58.43** | 47.44 | 26.62 | 31.48 | 49.61 | 33.93 |
| IMDB-10k | 32.82 | 32.53 | 29.47 | 12.69 | 23.31 | **48.51** | 35.45 |
| SST-10k | 58.83 | **68.08** | 65.03 | 27.32 | 67.7 | 59.49 | 44.76 |
| COLA | 69.32 | 82.06 | **84.79** | 44.59 | 42.05 | 68.46 | 71.23 |
| AGNEWS-10k | 220.03 | **221.59** | 186.54 | 95.6 | 202.22 | 208.1 | 179.2 |
| QQP-10k | 10.5 | 25.4 | 64.85 | 22.53 | 0.01 | **90.66** | 35.88 |
| MNLI-10k | 20.29 | 82.66 | 46.99 | 115.1 | 34.8 | **164.62** | 71.13 |
| YELP-10k | 19.27 | 33.91 | **39.9** | 8.13 | 10.12 | 33.49 | 25.33 |
| CHEMPROT | 499.58 | **646.59** | 613.5 | 603.17 | 601.67 | 619.9 | 446.36 |
| SCICITE | 164.7 | 171.53 | 163.86 | 88.84 | 108.6 | **190.12** | 152.38 |
| Mean | 92.13 | 118.64 | 118.04 | 90.89 | 93.2 | **134.96** | 91.2 |
| Average Rank | 5.08 | 2.62 | 2.85 | 5.38 | 5.31 | **2.31** | 4.46 |

Table 31: GM-10

| Tasks | STD++ | DA | WC | ReInit | R3F | REPINA$_I$ | REPINA$_{MLP}$ |
|---|---|---|---|---|---|---|---|
| RTE | 0.0 | 0.0 | 9.0 | 14.79 | 0.1 | **41.73** | 7.15 |
| MRPC | 15.09 | 24.73 | **33.48** | 26.9 | 13.53 | 17.29 | 3.31 |
| QNLI-10k | 6.99 | 9.65 | **16.49** | 3.13 | 7.1 | 11.6 | 12.37 |
| SCITAIL-10k | 5.46 | **14.88** | 14.11 | 4.35 | 6.74 | 10.75 | 7.91 |
| IMDB-10k | 6.7 | 6.83 | 7.81 | 1.7 | 4.09 | **11.92** | 6.44 |
| SST-10k | 15.31 | 19.3 | **21.21** | 4.19 | 18.37 | 16.18 | 9.32 |
| COLA | 19.72 | 24.21 | **28.76** | 11.5 | 9.88 | 19.34 | 20.15 |
| AGNEWS-10k | 65.88 | **67.33** | 54.52 | 18.78 | 60.41 | 62.36 | 44.45 |
| QQP-10k | 1.56 | 5.6 | 17.67 | 3.68 | 0.0 | **26.38** | 9.33 |
| MNLI-10k | 5.68 | 27.28 | 13.1 | 28.45 | 10.77 | **50.47** | 20.74 |
| YELP-10k | 3.28 | 7.42 | **11.0** | 0.95 | 1.88 | 7.14 | 4.95 |
| CHEMPROT | 302.28 | **429.53** | 333.57 | 356.82 | 332.11 | 389.25 | 226.06 |
| SCICITE | 48.68 | 54.7 | 56.06 | 18.57 | 29.54 | **59.65** | 45.86 |
| Mean | 38.2 | 53.19 | 47.44 | 37.99 | 38.04 | **55.7** | 32.16 |
| Average Rank | 5.08 | 2.77 | **2.31** | 5.31 | 5.46 | 2.46 | 4.62 |

Table 32: HM-10

## O  Walltime Analysis

**Walltime analysis:** STD++ uses a single forward and backward pass with simplest loss function and thus has the least training time. ReInit is a close second as it only differs in the initialization of the model. WC also uses a single forward and backward pass but is slower due to the regularization loss function computation. R3F and DA use two forward passes and two (effective) backward passes. Our method on the other hand use only one forward and backward pass. In addition to that we use only an extra forward pass of the pretrained model. Thus, our method is slower than STD++ , ReInit and WC and is faster than R3F and DA. Table 33 show the training time for all the methods. We observe that R3F consistently takes more time than all the methods. REPINA$_I$ runs faster than R3F and DA but slower than STD++ , WC and ReInit. REPINA$_{MLP}$ runs slower than REPINA$_I$ .

| Tasks | STD++ | DA | WC | ReInit | R3F | REPINA$_I$ | REPINA$_{MLP}$ |
|---|---|---|---|---|---|---|---|
| CHEMPROT | 584.01 | 1015.37 | 702.38 | 589.39 | 1415.28 | 826.09 | 1141.63 |
| MNLI-10k | 1506.74 | 2643.21 | 1723.52 | 1514.14 | 3746.99 | 1832.83 | 3022.97 |
| SCITAIL-10k | 1678.89 | 2962.31 | 1896.3 | 1684.06 | 4217.93 | 2418.12 | 3130.63 |
| MRPC | 162.09 | 258.92 | 214.94 | 157.42 | 362.83 | 226.59 | 299.13 |
| QNLI-10k | 1683.58 | 2966.56 | 1894.97 | 1681.34 | 4232.07 | 2414.78 | 3124.71 |
| QQP-10k | 1272.46 | 2194.95 | 1487.52 | 1276.92 | 3078.56 | 2307.62 | 2302.41 |
| SST | 5050.32 | 8537.84 | 5997.04 | 5080.05 | 11878.49 | 6019.1 | 9568.79 |
| COLA | 239.78 | 375.78 | 349.94 | 240.61 | 515.27 | 292.32 | 398.9 |
| SCICITE | 961.03 | 1746.23 | 1066.78 | 962.4 | 2838.07 | 1850.68 | 1835.6 |
| IMDB-10k | 1679.74 | 2975.29 | 1899.58 | 1692.33 | 4220.9 | 3149.3 | 3124.5 |
| YELP-10k | 1681.65 | 2975.09 | 1894.85 | 1686.21 | 4224.52 | 2426.62 | 3130.62 |
| RTE | 307.13 | 504.8 | 396.35 | 309.26 | 698.16 | 527.76 | 529.66 |
| AGNEWS-10k | 589.23 | 1046.27 | 675.37 | 595.06 | 1449.9 | 1125.27 | 1227.09 |
| Mean | 1338.2 | 2323.28 | 1553.81 | 1343.78 | 3298.38 | 1955.16 | 2525.9 |

Table 33: Training time for different methods

## P  Connection of GM and HM to parameter estimation error

Let the pseudo linear regression task on finetuned representations be defined by $w \in \mathbb{R}^d$ and the noisy labels observed on $z_i$'s be $y_i = z_i^T w + \epsilon_i$ where $\epsilon_i$'s are the gaussian noise centered around $\mathbf{0}$ with identity covariance matrix. If $\hat{w}$ is the least square minimizer (same as log-likelihood maximizer), then $\hat{w} = w + N\left(0, G^{-1}\right)$

GM corresponds to minimizing the confidence ellipsoid corresponding to the error $\hat{w} - w$. HM corresponds to minimizing the expected $\ell_2^2$ norm of the error vector $\hat{w} - w$. Derivation of the $\hat{w}$ and the explanation can be found in Madan et al. (2019).