# OpenReview forum: "Representation Projection Invariance Mitigates Representation Collapse"
_EMNLP/2023/Conference — EMNLP 2023 Findings_

### Official Review · Reviewer_1NmN · 2023-07-24

**Soundness:** 4

**Excitement:**

3: Ambivalent: It has merits (e.g., it reports state-of-the-art results, the idea is nice), but there are key weaknesses (e.g., it describes incremental work), and it can significantly benefit from another round of revision. However, I won't object to accepting it if my co-reviewers champion it.

**Paper Topic And Main Contributions:**

The paper introduces a novel regularization method, REPINA, designed to address representation collapse during the fine-tuning of pre-trained language models. The study evaluates REPINA against five comparable baselines across a diverse set of 13 language understanding tasks, and find REPINA consistantly outperform other baselines on most of them.

**Questions For The Authors:**

- In line 354, does the "fail run" filter applied to all of the baselines in your comparsons?
- In terms of REPINA's performance, it appears to lag behind the other baselines in certain tasks. Do you have any insights regarding the specific reasons behind these shortcomings?

**Reasons To Accept:**

- The proposed Idea is clear and effective, leading to consistant improvement in fine-tuning
- The paper is well written
- The experiments and analysis are comprehansive


**Reasons To Reject:**

- The paper only perform with BERT-large model, leaving uncertainty regarding the applicability of the proposed approach to other pre-trained model architectures.


**Reproducibility:**

4: Could mostly reproduce the results, but there may be some variation because of sample variance or minor variations in their interpretation of the protocol or method.

**Reviewer Confidence:**

2: Willing to defend my evaluation, but it is fairly likely that I missed some details, didn't understand some central points, or can't be sure about the novelty of the work.

---

### Official Review · Reviewer_DTwA · 2023-08-08

**Soundness:** 3

**Excitement:**

4: Strong: This paper deepens the understanding of some phenomenon or lowers the barriers to an existing research direction.

**Missing References:**

Zheng, Bo, et al. "Consistency regularization for cross-lingual fine-tuning." arXiv preprint arXiv:2106.08226 (2021) might be relevant in the context of data augmentation based regularization (L215)
Bahri, Dara, Hossein Mobahi, and Yi Tay. "Sharpness-aware minimization improves language model generalization." arXiv preprint arXiv:2110.08529 (2021) might be applicable for general purpose fine-tuning approaches.


**Paper Topic And Main Contributions:**

The paper presents a new regularization method: representation projection invariance regularization dubbed REPINA. Specifically, the authors

1. Present a new regularization term that is based on the distance between the pre-trained representations and a transformed version of the fine-tuned representations, where the class of applicable transformations determines the strength of the regularization.
2. Draw a connection between their proposed method and a multi-task setup with infinite pseudo tasks.
3. Demonstrate on 13 tasks the effectiveness of their proposed approach compared to other baselines for 10 / 13 tasks
4. Show that REPINA allows for higher run stability, OOD performance, Robustness to label noise and better performance on the few-shot fine-tuning setup
5. Demonstrate that the proposed method shows less representation collapse when measured using probing experiments
6. Propose a new method to quantify dimensional collapse based on the eigenvalues of the Gram matrix, showing that REPINA still shows least amount of representational collapse compared to other metrics.

**Questions For The Authors:**

1. Would it be possible to also report the OOD results for RP_{MLP} in Table 3 ?
2. Would it be possible to also consider the motivating set of linear projections as a part of the experiments ? Since it is a linear projection, the solution to the inner minimum can be obtained analytically at each step of the optimization process.

**Reasons To Accept:**

1. The proposed method, while simple (especially REPINA_{I}) demonstrates strong performance and low levels of representational collapse compared to other baseline approaches
2. The experiments are quite comprehensive, across a diverse number of datasets and compared against a wide variety of baselines
3. The connection to multi task learning is quite intriguing, and merits additional exploration
4. The ablation studies w.r.t OOD robustness, robustness to label perturbation as well as the experiments measuring representational collapse underscore the ability of the proposed method in preserving the pre-trained models representational space.

**Reasons To Reject:**

1. The setup of the MLP formulation is a bit underspecified: is the min actually solving an inner optimization problem (so at every training step, the MLP minimized (via multiple steps of gradient descent on the MLP's parameters), or is it a first order approximation (so only one step of minimization for each training step). Is there a performance delta between the two approaches ? For the first one, the wall-clock time would substantially increase (because there would be multiple gradient descent steps on the MLP's parameters per optimization step).

2. Some of the baselines omit stronger counter-parts. For example, weight consolidation does not consider the EWC variant that upweighs feature importance based on the the approximate fisher information (eg: see [1]). That has been shown to perform better than vanilla EWC. Likewise, Sharpness Aware Minimization techniques have been demonstrated to perform better than R3F and R4F ([2]). Hence those should be used to establish the baselines, in my opinion.

3. The proposed Diversity of Fine-tuned representations methodology for understanding representational collapse requires additional explorations. Specifically, correlation studies between the proposed method and linear probes, as well as stability while using across different datasets needs to be quantified before the metric can be adopted as another way for checking for representational collapse. In addition to that, the metric (to my understanding) is analogous to looking at the variance of datapoints along the principle components of the representations. Thus, there are cases where this would not be able to detect a representational collapse (eg: say the collapse causes the space to be uniformly spread along a sphere, as opposed to occupying the entire space. In that case, there is a collapse, but this would not be detected by the proposed method). At the very least, the stability across different datasets and correlation metrics would inspire some level of confidence for the proposed method.


[1] Thorne, James, and Andreas Vlachos. "Elastic weight consolidation for better bias inoculation." arXiv preprint arXiv:2004.14366 (2020).
[2] Bahri, Dara, Hossein Mobahi, and Yi Tay. "Sharpness-aware minimization improves language model generalization." arXiv preprint arXiv:2110.08529 (2021).

**Reproducibility:**

4: Could mostly reproduce the results, but there may be some variation because of sample variance or minor variations in their interpretation of the protocol or method.

**Reviewer Confidence:**

3: Pretty sure, but there's a chance I missed something. Although I have a good feel for this area in general, I did not carefully check the paper's details, e.g., the math, experimental design, or novelty.

**Typos Grammar Style And Presentation Improvements:**

L130: we bias "fine-tuned representations to be linear transformations of the fine-tuned representations -> we bias pre-trained representations to be linear transformations of the fine-tuned representations ?
L133: Thus, regularization loss in case of Fig 2a would be zero -> Thus, regularization loss in case of Fig 2b would be zero ?

---

### Official Review · Reviewer_4e8F · 2023-08-10

**Soundness:** 2

**Excitement:**

3: Ambivalent: It has merits (e.g., it reports state-of-the-art results, the idea is nice), but there are key weaknesses (e.g., it describes incremental work), and it can significantly benefit from another round of revision. However, I won't object to accepting it if my co-reviewers champion it.

**Paper Topic And Main Contributions:**

This paper addresses the issue of representation degradation (representation collapse) that can occur during fine-tuning of pre-trained language models in NLP. It introduces a novel regularization method called Representation Projection Invariance (REPINA) to mitigate representation collapse by discouraging undesirable changes in representations while fine-tuning. The study empirically compares the proposed method with five comparable baselines across 13 language understanding tasks, including the GLUE benchmark and additional datasets. The results demonstrate REPINA's consistent outperformance of other baselines on most tasks, both in-domain and out-of-distribution. The paper also presents the extension of previous studies on representation collapse by introducing metrics to quantify it.



**Questions For The Authors:**

How do you envision extending the insights gained from REPINA to other aspects of fine-tuning, such as transfer learning, multi-task learning, or domain adaptation, where representation collapse could also play a role?



**Reasons To Accept:**

1、The paper addresses a significant challenge in NLP – representation collapse during fine-tuning – and presents a novel solution, REPINA. This issue is particularly relevant given the widespread use of pre-trained language models.

2、The extension of previous studies by introducing metrics to quantify representation collapse enhances the paper's contribution to the field. This added insight provides a valuable framework for analyzing and addressing this issue.

The demonstrated effectiveness of REPINA in in-domain, out-of-distribution, few-shot settings, and robustness to label perturbation highlights its versatility and potential applicability across a range of NLU scenarios.

**Reasons To Reject:**

1、The paper mentions that REPINA consistently outperforms baselines on most tasks. However, it could provide a more detailed comparison, including specific performance metrics and a deeper analysis of why REPINA performs better.

2、The author should explore more about the applicability of the method in this paper to prompt tuning settings and LLMs.

**Reproducibility:**

3: Could reproduce the results with some difficulty. The settings of parameters are underspecified or subjectively determined; the training/evaluation data are not widely available.

**Reviewer Confidence:**

4: Quite sure. I tried to check the important points carefully. It's unlikely, though conceivable, that I missed something that should affect my ratings.

---

### Official Review · Reviewer_E5eQ · 2023-08-11

**Soundness:** 4

**Excitement:**

3: Ambivalent: It has merits (e.g., it reports state-of-the-art results, the idea is nice), but there are key weaknesses (e.g., it describes incremental work), and it can significantly benefit from another round of revision. However, I won't object to accepting it if my co-reviewers champion it.

**Missing References:**

[1] Hu, Edward J., Yelong Shen, Phillip Wallis, Zeyuan Allen-Zhu, Yuanzhi Li, Shean Wang, Lu Wang, and Weizhu Chen. "Lora: Low-rank adaptation of large language models." arXiv preprint arXiv:2106.09685 (2021).

**Paper Topic And Main Contributions:**

The paper addresses the critical issue of representation collapse during fine-tuning of language models for downstream tasks. The authors propose a regularization-based approach called REPINA to maintain the expressivity within latent representations throughout the fine-tuning process. The evaluation encompasses 13 datasets, along with comparisons against five distinct baseline methods.

**Questions For The Authors:**

* Could the authors provide the eigenvalues of the presentations for both unseen and seen tasks in the pre-trained model? It would be beneficial if Figure 5 could also illustrate the anticipated eigenvalues.

* Is there empirical validation supporting the selection of layer values as 5, 10, or 20?

* What is the mean performance (accuracy) for Table 5?

**Reasons To Accept:**

* One commendable aspect of the paper is its consistent achievement of state-of-the-art performance across multiple datasets. This is a testament to the efficacy of the proposed REPINA methodology.

* The authors' inclusion of comprehensive ablation studies contributes significantly to the clarity and depth of their findings. These studies enhance our understanding of the tangible impact REPINA can have on the fine-tuning process.

* The paper underscores its merit by demonstrating the superiority of REPINA over other baseline methods in various robustness scenarios, including out-of-distribution settings, label perturbations, and few-shot fine-tuning. This underlines the versatility of the proposed approach.

* The paper is well written.

**Reasons To Reject:**

While the paper demonstrates impactful contributions to the field, there are certain aspects that merit further attention:

* One notable omission is the absence of an essential baseline, namely LoRA for LMs (Low-Rank Adaptation) [1], which could offer a meaningful point of comparison.
* Additionally, expanding the experimentation to include diverse models beyond BERT or RoBERTa, such as T5 or GPT-2, would strengthen the paper's conclusions.

Note: While it is acceptable to concentrate ablations or analyses on one model, I recommend diversifying the model selection and adding similar tables as Table 1 for different models. Including experiments across a wider array of models would provide a more comprehensive perspective on the methodology's applicability.

**Reproducibility:**

5: Could easily reproduce the results.

**Reviewer Confidence:**

4: Quite sure. I tried to check the important points carefully. It's unlikely, though conceivable, that I missed something that should affect my ratings.

---

### Official Review · Reviewer_HLED · 2023-08-12

**Soundness:** 3

**Excitement:**

3: Ambivalent: It has merits (e.g., it reports state-of-the-art results, the idea is nice), but there are key weaknesses (e.g., it describes incremental work), and it can significantly benefit from another round of revision. However, I won't object to accepting it if my co-reviewers champion it.

**Missing References:**

* (2021) Training Neural Networks with Fixed Sparse Masks
* (2021) LORA: LOW-RANK ADAPTATION OF LARGE LAN- GUAGE MODELS

**Paper Topic And Main Contributions:**

This paper proposes a new regularization method called Representation Projection Invariance (REPINA) to mitigate representation collapse during fine-tuning of pretrained language models. The key idea is to regularize the fine-tuned representations to be invariant to projections by a chosen set of functions. The paper presents two versions of REPINA – REPINAI uses an identity function for projection and REPINAMLP uses a multilayer perceptron, which allows applying stronger or milder regularization.

**Questions For The Authors:**

1. line 130-138: I don’t understand the logic of “regularization loss would be zero … however … regularization loss would be high”. The writing could be improved.
2. For the OOD experiments in Section 5.3, using a bar chart to show generalization gap as in Hendrycks et al. (2020) would make the results clearer. The current table format is difficult to parse for assessing OOD robustness.

**Reasons To Accept:**

* This paper proposes a novel regularization approach targeting representation collapse during fine-tuning, complementing existing techniques that regularize model weights/gradients.
* The experiments are well-structured, extensive, and solid, covering OOD evaluation, label perturbation, probing representation collapse, representation diversity, etc.
* The paper is generally well-written and easy to follow.

**Reasons To Reject:**

* This paper only compares the proposed methods with models published before 2021. It lacks discussion of recent parameter-efficient fine-tuning strategies like fixed sparse masking and low-rank adaptation.
* Section 2 claims REPINAI allows stronger regularization than REPINAMLP but provides no direct experimental evidence supporting this intuition.


**Reproducibility:**

4: Could mostly reproduce the results, but there may be some variation because of sample variance or minor variations in their interpretation of the protocol or method.

**Reviewer Confidence:**

4: Quite sure. I tried to check the important points carefully. It's unlikely, though conceivable, that I missed something that should affect my ratings.

---

### Meta-Review · Area_Chair_V4Za · 2023-09-14

**Recommendation:** 2

**Metareview:**

This paper proposes a novel regularization approach targeting representation collapse during fine-tuning, complementing existing techniques that regularize model weights/gradients. The experiments are well-structured, extensive, and solid, covering OOD evaluation, label perturbation, probing representation collapse, representation diversity, etc. The paper is generally well-written and easy to follow. However, this paper only compares the proposed methods with models published before 2021. It lacks discussion of recent parameter-efficient fine-tuning strategies like fixed sparse masking and low-rank adaptation. Besides, the paper mentions that REPINA consistently outperforms baselines on most tasks. However, it could provide a more detailed comparison, including specific performance metrics and a deeper analysis of why REPINA performs better.

---

### Decision · Program_Chairs · 2023-10-07

**Decision:**

Accept-Findings

**Comment:**

This paper proposes a novel regularization approach targeting representation collapse during fine-tuning, complementing existing techniques that regularize model weights/gradients. The experiments are well-structured, extensive, and solid, covering OOD evaluation, label perturbation, probing representation collapse, representation diversity, etc. The paper is generally well-written and easy to follow. However, this paper only compares the proposed methods with models published before 2021. It lacks discussion of recent parameter-efficient fine-tuning strategies like fixed sparse masking and low-rank adaptation. Besides, the paper mentions that REPINA consistently outperforms baselines on most tasks. However, it could provide a more detailed comparison, including specific performance metrics and a deeper analysis of why REPINA performs better.